# Neuroinvasion and anosmia are independent phenomena upon infection with SARS-CoV-2 and its variants

Guilherme Dias de Melo [1], Victoire Perraud [1,14], Flavio Alvarez [2,3,14], Alba Vieites-Prado [4,14], Seonhee Kim [1], Lauriane Kergoat [1], Anthony Coleon [1], Bettina Salome Trüeb[5], Magali Tichit[6], Aurèle Piazza[7], Agnès Thierry[7], David Hardy[6], Nicolas Wolff[2], Sandie Munier[8], Romain Koszul [7], Etienne Simon-Lorière [9], Volker Thiel [10], Marc Lecuit [11,12], Pierre-Marie Lledo[13], Nicolas Renier [4], Florence Larrous [1,15] & Hervé Bourhy [1,15] ✉

Anosmia was identified as a hallmark of COVID-19 early in the pandemic, however, with the emergence of variants of concern, the clinical profile induced by SARS-CoV-2 infection has changed, with anosmia being less frequent. Here, we assessed the clinical, olfactory and neuroinflammatory conditions of golden hamsters infected with the original Wuhan SARS-CoV-2 strain, its isogenic ORF7-deletion mutant and three variants: Gamma, Delta, and Omicron/BA.1. We show that infected animals develop a variant-dependent clinical disease including anosmia, and that the ORF7 of SARS-CoV-2 contributes to the induction of olfactory dysfunction. Conversely, all SARS-CoV-2 variants are neuroinvasive, regardless of the clinical presentation they induce. Taken together, this confirms that neuroinvasion and anosmia are independent phenomena upon SARS-CoV-2 infection. Using newly generated nanoluciferase-expressing SARS-CoV-2, we validate the olfactory pathway as a major entry point into the brain in vivo and demonstrate in vitro that SARS-CoV-2 travels retrogradely and anterogradely along axons in microfluidic neuron-epithelial networks.

The COVID-19 pandemic remains a major global public health problem. Since the beginning of the pandemic in December 2019, more than 760 million cases have been confirmed, all SARS-CoV-2 variants combined[1]. The original SARS-CoV-2 (Wuhan) gave rise to different variants of concern (VoCs) in the first year of the pandemic (Alpha/B.1.1.7, Beta/B.1.351, Gamma/P.1), which were almost totally replaced by the VoC Delta (B1.617.2, AY*) in 2020, by the VoC Omicron and its lineages (e.g. BA.1, BA.2 and BA.5) in 2021-2022 and recombinants (XBB*)[1].

SARS-CoV-2 infects cells of the upper and lower airways, and COVID-19 manifests by a multitude of respiratory and extra-respiratory symptoms, including neurological manifestations, ranging from headache and dizziness to anosmia, ageusia and even stroke[2–4]. The neuropathology of COVID-19 is currently considered to be a consequence of inflammation and hypoxia, rather than direct viral invasion into the CNS[5,6]. However, with the emergence of the VoCs, and the increasing rate of vaccination or previous infection, the symptomatology of COVID-19 has changed. In this context, the clinical picture induced by the VoC Omicron/BA.1 is less severe or sometimes asymptomatic, with a higher rate of upper airways involvement, sparing the olfactory mucosa and consequently a lower incidence of anosmia, initially considered a hallmark of COVID-19[7–13].

Previously, we determined that the infection of olfactory sensory neurons and loss of cilia in the olfactory mucosa caused by SARS-CoV-2 Wuhan in humans and Syrian hamsters were associated to olfaction loss, as well as to local inflammation and neuroinflammation in the olfactory bulbs, attesting that the golden hamster is a relevant model to study the pathogenesis of SARS-CoV-2 infection[14–16]. Other authors reported persistent microgliosis in the olfactory bulbs of infected hamsters[17], and differences in the neuroinvasiveness and neurovirulence of VoCs have been described[18]. Additionally, other authors have attempted to compare VoCs pathogenicity in hamsters[19–21], but studies relating SARS-CoV-2 brain invasion to neurological symptoms are lacking. In addition to the pathogenicity variation due to mutations in the spike of the variants, other viral proteins may play a role in anosmia and local inflammation, including ORF7, which has been shown to interfere with the host's innate immunity[22–24], with cell adhesion in the olfactory mucosa[25], and with olfactory receptors[26].

Here, we show that SARS-CoV-2 Wuhan and the VoCs Gamma, Delta and Omicron/BA.1 are all capable of invading the brain of Syrian hamsters and of eliciting a tissue-specific inflammatory response. Using reverse genetics-generated bioluminescent viruses, we demonstrate that SARS-CoV-2 infects the olfactory bulbs, but the clinical profile, including the olfactory performance, is highly dependent on the variant. Further, deletion of the ORF7ab sequence in the ancestral virus (Wuhan) reduces the incidence of olfaction loss without affecting the clinical picture nor the neuroinvasiviness via the olfactory bulbs. Accordingly, this work validates that SARS-CoV-2 may travel inside axons and the olfactory pathway as the main entry route by SARS-CoV-2 into the brain and corroborates the neurotropic potential of SARS-CoV-2 variants. Neuroinvasion and anosmia are therefore independent phenomena upon SARS-CoV-2 infection.

## Results

### SARS-CoV-2 induces a clinical disease in hamsters, with a VoC-related severity difference

We first investigated the differences in the clinical picture induced by different SARS-CoV-2 VoCs in comparison with the ancestral virus (Wuhan). We first tested the in vitro growth curves of these viruses in Vero-E6 cells, and found no significant differences (Supplementary Fig. 1A). Then, male golden hamsters were inoculated intranasally with $6 \times 10^4$ PFU of SARS-CoV-2 Wuhan, or the VoCs Gamma, Delta or Omicron/BA.1 and followed them up for 4 days post-infection (dpi). All SARS-CoV-2-infected animals presented a progressive loss of weight; however, a variant effect was observed (Kruskal-Wallis $P < 0.0001$, Fig. 1A, B), with SARS-CoV-2 Wuhan-infected animals presenting the most intense median weight loss (15.8%, interquartile range 'IQR' 3.9%), followed by Gamma-infected animals (11.0%, IQR 2.6%), Delta-infected animals (9.4%, IQR 3.1%) and Omicron/BA.1-infected animals (1.9%, IQR 2.5%). Non-specific sickness-related clinical signs (ruffled fur, slow movement, apathy) followed the same pattern (Kruskal-Wallis $P < 0.0001$, Fig. 1D, E), with SARS-CoV-2 Wuhan-infected animals presenting the worse clinical picture, followed by Gamma- and Delta-infected animals, while Omicron/BA.1-infected animals presented a delayed manifestation of mild signs, clinically observable at 4 dpi only.

Olfaction loss is a typical clinical sign of SARS-CoV-2 Wuhan-infected hamsters[14], however olfactory deficit differed according to the different VoCs (Chi-square for trend $P < 0.0001$): 62.5% (5/8) of SARS-CoV-2 Wuhan-infected animals presented loss of olfaction (Log-rank test compared to the mock $P = 0.0012$), only 12.5% (1/8) of Gamma-infected animals lost olfaction completely with 62.5% (5/8) presenting an impaired olfactory performance (i.e. longer time to find the hidden cereals) (Log-rank test compared to the mock $P < 0.0001$). In contrast, none of the Delta- and Omicron/BA.1-infected animals presented signs of olfactory impairment (Fig. 1G, H). All animals found the visible food during the control test, demonstrating that no sickness behavior, visual impairment, or

locomotor deficit was responsible for the delay in finding the hidden food.

Several mutations on the spike distinguish the different VoCs. Furthermore, some VoC isolates also present deletions in the ORF7 sequence[27–30], including the Delta isolate used in the present study (deletion spanning positions 27506 to 27540 of the reference SARS-CoV-2 Wuhan genome, which generates a stop codon; Supplementary Fig. 1D). Since a possible link between ORF7b and olfaction loss has been proposed[25], we constructed a recombinant SARS-CoV-2 based on the CoV-2/W backbone, where the ORF7ab sequence was replaced by that of GFP (SARS-CoV-2 Wuhan/ΔORF7ab) (Supplementary Fig. 4A). Remarkably, the clinical profile exhibited by Wuhan/ΔORF7ab-infected hamsters was similar to the one caused by the wild-type SARS-CoV-2 Wuhan (Fig. 1A–F), showing that ORF7ab is not essential for viral infection and replication. However, the olfaction loss incidence decreased significantly; only 25% (2/8) of the infected animals presented signs of anosmia, in contrast to the 62.5% (5/8) observed in CoV-2/SARS-COV-2 Wuhan-infected hamsters (Fig. 1G, H). Despite this reduction in anosmia incidence, Wuhan/ΔORF7ab was still detected in the airways and in the olfactory bulbs of infected animals, with even higher titers (Fig. 2A, B).

Regarding lung pathology, as for the other clinical parameters, lung enlargement and lung weight-to-body weight (LW/BW) ratio were significantly higher in SARS-COV-2 Wuhan- and in Wuhan/ΔORF7ab-infected animals; Gamma- and Delta-infected animals presented intermediate values, whereas Omicron/BA.1-infected animals' values were close to those of the ones of mock-infected animals (Kruskal-Wallis $P < 0.0001$, Fig. 1C, F). Likewise, the histopathological findings in the lungs followed the same tendency: congestion, edema, mononuclear cells infiltration, thickening of the alveolar walls and bronchiolar epithelium desquamation were observed in all infected animals (Supplementary Fig. 2A), along with a diffuse SARS-CoV-2 nucleocapsid staining (Supplementary Fig. 2B), with more severe alterations observed in the lungs of SARS-COV-2 Wuhan-infected animals than in Gamma-, Delta-, and Omicron/BA.1-infected hamsters.

In a multivariate analysis of clinical parameters recorded at 4 dpi, the two-first principal components explained 91.9% of sample variability. Body weight loaded negatively to PC1 (principal component) and negatively correlated with the clinical score and the LW/BW ratio, whereas the olfactory performance loaded positively to PC2 (Supplementary Fig. 3A). In the principal component analysis (PCA) plot regarding tissue-related inflammation, the difference among groups was marked: mock and Omicron/BA.1-infected animals loaded homogeneously and relatively close, followed by the Delta-infected animals. SARS-COV-2 Wuhan-, Wuhan/ΔORF7ab-, and Gamma-infected animals loaded in a more dispersed way, far away from the other VoCs, effect of the olfactory performance deficit and clinical severity (Supplementary Fig. 3C).

### SARS-CoV-2-infected airways respond to the infection regardless of the viral variants

Next, we measured the viral titer and RNA loads in the upper airways (nasal turbinates), and in the lower airways (lung). Infectious viruses were detected in the nasal turbinates and in the lungs of all infected hamsters regardless of the VoC, however, a variant effect was observed, with the highest values for the Gamma-infected animals, and the lowest values for the Omicron/BA.1-infected animals (Kruskal-Wallis $P < 0.0001$, Fig. 2A). Genomic and sub-genomic SARS-CoV-2 RNA were detected equally in the lungs of all infected animals; but differently in the nasal turbinates, regardless of the positivity of all samples, Delta-infected animals presented the highest viral load (Kruskal-Wallis $P = 0.0023$, Fig. 2B).

Both the upper and lower airways responded to the infection by all VoCs (Fig. 2C, D), but with a tissue-specific inflammatory signature: in the lungs, *Mx2, Il-6, Cxcl10* and *Il-10* were upregulated for all VoCs,

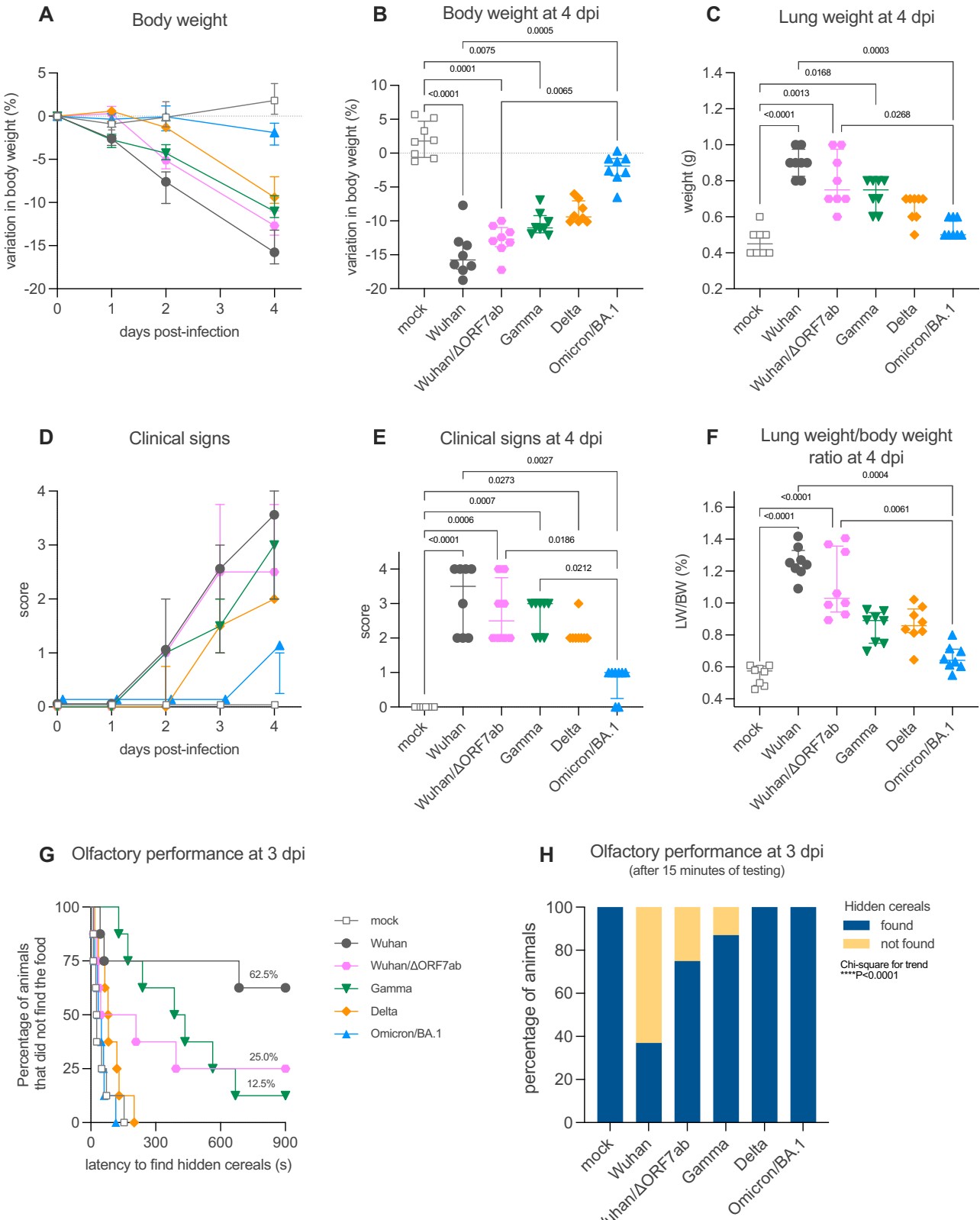

Fig. 1 | Clinical profile of hamsters infected with SARS-CoV-2 original virus (Wuhan), the recombinant Wuhan/ΔORF7ab or the variants of concern (VoC) Gamma, Delta, and Omicron/BA.1. A, B Body weight variation over four days post-infection. C Lung weight measured at 4 dpi. D, E Clinical score over four days post-infection. The clinical score is based on a cumulative 0–4 scale: ruffled fur; slow movements; apathy; and absence of exploration activity. F Lung weight-to-body weight ratio measured at 4 dpi. A–F Horizontal lines indicate median and the

interquartile range ($n = 8$/group). B, C, E, F Kruskal-Wallis test followed by the Dunn's multiple comparisons test (the adjusted $p$ value is indicated when significant). G, H Olfactory performance measured at 3 days post-infection (dpi). The olfaction test is based on the hidden (buried) food finding test. Curves represent the olfactory performance of animals during the test (G) and bars represent the final results (H) ($n = 8$/group). Chi-square test for trend (the adjusted p value is indicated when significant). See Supplementary Figs. 1, 2.

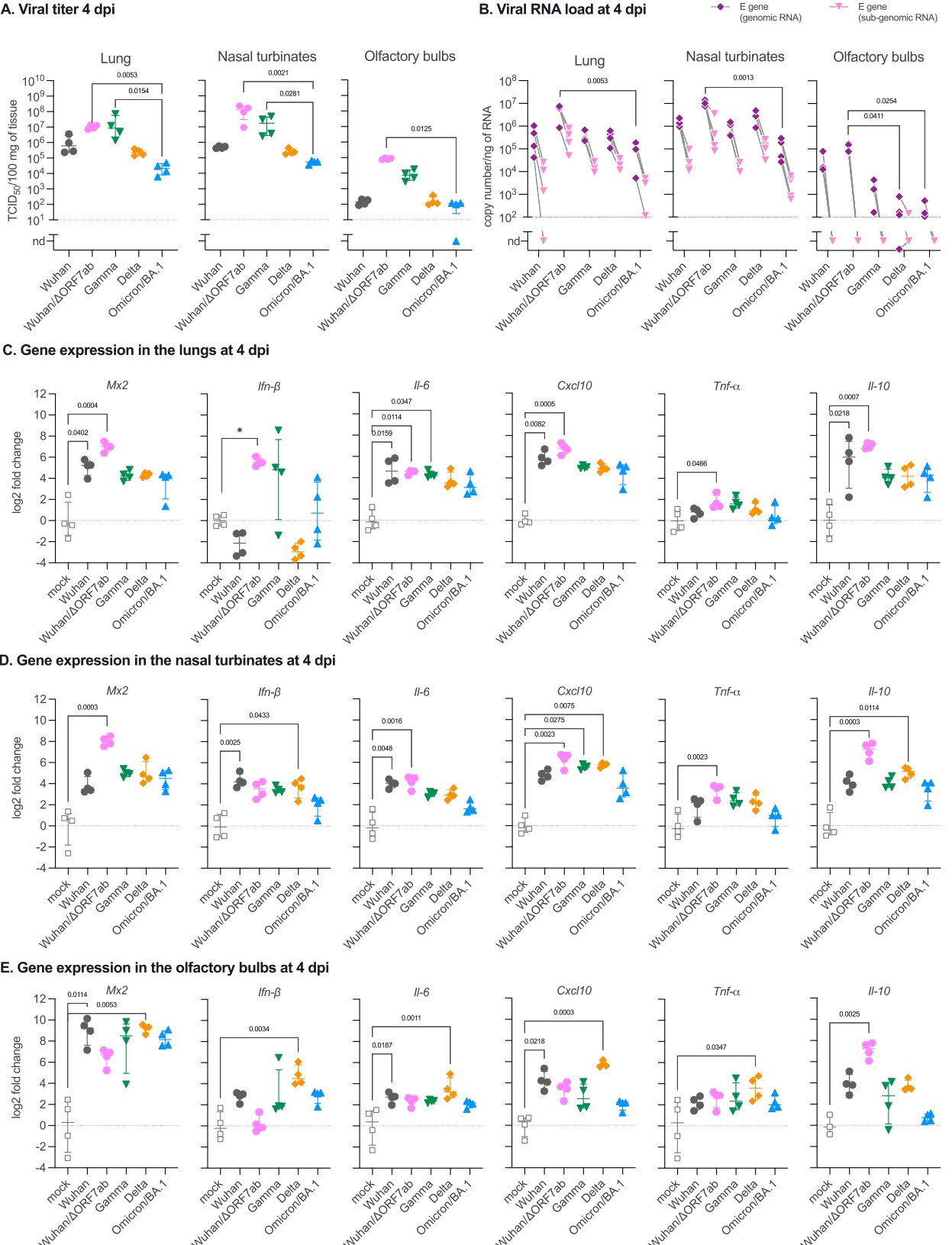

**A. Viral titer 4 dpi**

**B. Viral RNA load at 4 dpi**

**C. Gene expression in the lungs at 4 dpi**

**D. Gene expression in the nasal turbinates at 4 dpi**

**E. Gene expression in the olfactory bulbs at 4 dpi**

and the highest in SARS-COV-2 Wuhan-infected animals (Fig. 2C). In the nasal turbinates, the gene expression of *Ifn-β* and *Il-6* presented the highest values in SARS-COV-2 Wuhan-infected animals and the lowest in Omicron/BA.1-infected animals. *Mx2*, *Cxcl10* and *Il-10* expression were the highest in Delta-infected animals while Omicron/BA.1-infected animals showed the lowest levels (Fig. 2D). Interestingly, Wuhan/

ΔORF7ab infection induced a different lung inflammatory signature compared to Wuhan or the other VoCs, represented by a higher expression of *Mx2* and *Ifn-β* (Fig. 2C). In the nasal turbinates, Wuhan/ΔORF7ab infection also induced a higher expression of *Mx2*, but also *Il-6, Cxcl10, Tnf-α* and *Il-10* (Fig. 2D). To better appreciate the tissue- *vs.* virus-related gene expression, we performed a multivariate analysis of

**Fig. 2 | Virologic assessment and gene expression of selected immune mediators in different tissues of hamsters infected with SARS-CoV-2 original virus (Wuhan), the recombinant Wuhan/ΔORF7ab or the variants of concern (VoC) Gamma, Delta, and Omicron/BA.1. A** Infectious viral titers in the lung, nasal turbinates, and olfactory bulbs at 4 days post-infection (dpi) expressed as TCID$_{50}$ per 100 mg of tissue. Horizontal lines indicate median and the interquartile range ($n = 4$/group). **B** SARS-CoV-2 viral RNA load detected in the lung, nasal turbinates, and olfactory bulbs at 4 dpi. Genomic and sub-genomic viral RNA were assessed based on the E gene sequence. Horizontal lines indicate median and the interquartile range. Gray lines connect symbols from the same animals ($n = 4$/group). Gene expression values in the lung (**C**), nasal turbinates (**D**) and olfactory bulbs (**E**) of *Mx2, Ifn-β, Il-6, Cxcl10, Tnf-α* and *Il-10* at 4 dpi. **A–E** Horizontal lines indicate median and the interquartile range ($n = 4$/group). Kruskal-Wallis test followed by the Dunn's multiple comparisons test (the adjusted *p* value is indicated when significant). nd: not detected. See Supplementary Figs. 1, 2.

these data. Interestingly, the two-first principal components explained 79.5% of sample variability (Supplementary Fig. 3B). In the PCA plots, the nasal turbinates of all infected hamsters were loaded in proximity to each other (Supplementary Fig. 3D), whereas the lungs were separated by an important effect of *Il-6, Cxcl10* and *Il-10* (Supplementary Fig. 3D).

### SARS-CoV-2 neuroinvasion and neuroinflammation in the olfactory bulbs

After establishing the clinical and inflammatory profile of the infected animals, we aimed to assess the effect of infection on the olfactory bulbs. Remarkably, even if the olfactory performance differed according to the VoCs (Fig. 1G, H), positive viral titers were detected in the olfactory bulbs of animals from all infected groups, with Wuhan/ΔORF7ab and Gamma-infected animals presenting the highest titer at 4 dpi (Kruskal-Wallis *P* = 0.0096, Fig. 2A). These findings were corroborated by the detection of genomic viral RNA in the olfactory bulbs of animals from all infected groups as well, but viral RNA load was higher in SARS-COV-2 Wuhan-infected animals (Kruskal-Wallis *P* = 0.0104, Fig. 2B). Conversely, in these samples, the sub-genomic RNA was below the detection limit.

The olfactory bulbs presented an intriguing inflammatory profile. The antiviral *Mx2* gene, along with the inflammatory genes *Ifn-β, Il-6* and *Cxcl10* were highly upregulated in the olfactory bulbs of all infected hamsters (Fig. 2E), regardless of their olfactory performance in the food-finding test (Fig. 1G, H), yet *Il-10* expression was highly upregulated in the olfactory bulbs of SARS-COV-2 Wuhan/ΔORF7ab-infected animals (compared to the mock). Unexpectedly, the gene expression of these selected targets tended to be higher in the olfactory bulbs of Delta-infected animals (Fig. 2E). In a multivariate analysis for tissue- and virus-related gene expression, the olfactory bulbs from SARS-COV-2 Wuhan-infected animals tended to load in proximity to the corresponding nasal turbinates (effect of *Il-6, Cxcl10* and *Il-10*), whereas olfactory bulbs from Gamma-, Delta- and Omicron/BA.1-infected animals loaded separately, possibly reflecting the impact of *Mx2, Tnf-α* and *Ifn-β* differential expression (Supplementary Fig. 3D).

Having detected the virus in the olfactory bulbs using virologic and molecular techniques, we aimed to visualize the infection. We examined the SARS-CoV-2 Wuhan distribution in the whole brain by combining whole-head tissue clearing with light sheet microscopy imaging using iDISCO+[31]. At 4 dpi, all Wuhan-infected hamsters displayed a diffuse viral distribution in the nasal cavity: nasal turbinates and olfactory epithelium (Fig. 3A and Supplementary Movie 1). Along the same lines, all SARS-COV-2 Wuhan-infected animals presented SARS-CoV-2 in the olfactory bulbs in sparse neurons, localized in the proximity of the olfactory nerve entry point (Fig. 3B and Supplementary Movie 1). Using this technique, no infected cells were observed in other areas of the brain and no alteration in the vascular network was detected.

We next used a complementary method to visualize the infection of the olfactory bulbs. To this end, we generated recombinant viruses that express the nanoluciferase by reverse genetics (Wuhan_nLuc, Wuhan/ΔORF7ab_nLuc, and Delta_nLuc; Supplementary Fig. 1B, C and Supplementary Fig. 4A). The disease profile induced by these three novel recombinant viruses was similar to the profile induced by the wild-type parental viruses (Fig. 3C). In vivo imaging of infected

hamsters was performed at 4 dpi using fluorofurimazine (FFz) as substrate. Very low positive signals were observed in the nasal cavity region, possibly due to the thickness of the skin and bones that could block the light emission, but significantly higher in the Wuhan_nLuc than in the Wuhan/ΔORF7ab_nLuc the Delta_nLuc (Kruskal-Wallis *P* = 0.0107, Supplementary Fig. 4BC). More sensitive results were obtained during the ex vivo imaging. After euthanasia, the lungs and brains were collected and imaged. Positive signals in the lungs were obtained at the same intensity for the three viruses (Fig. 3D), and presented a multi-focal/diffuse distribution (Fig. 3E and Supplementary Fig. 4D). The brains were imaged in the ventral position in order to better expose the olfactory bulbs. Intense luminescent signals were recorded from the brains of hamsters infected with all viruses; when placing the ROI (region of interest) exclusively in the olfactory bulbs, intensity was higher in Wuhan_nLuc compared to Delta_nLuc; conversely, when considering the whole brain ventral area, intensity was higher in Delta_nLuc. Wuhan/ΔORF7ab_nLuc induced intermediate values (Fig. 3C and Supplementary Fig. 4D), which might reflect the higher replication rate detected in the olfactory bulbs infected Wuhan (Fig. 2B). Despite intensity differences, the olfactory bulbs were the major infected structure in the brain, and sagittal views of the brain indicate that the origin of the luminescent signal is the ventral part of the olfactory bulbs, which is situated above the olfactory epithelium in the nasal cavity (Fig. 3E and Supplementary Fig. 4E).

### Anosmia is not associated with viral load or infection of the olfactory bulbs

All tested SARS-CoV-2 variants were found to be able to invade the CNS and infect the olfactory bulbs, but the incidence of olfactory dysfunctions varied according to the VoC, with hamsters infected with Delta and Omicron/BA.1 presenting no signs of anosmia (Fig. 1G, H). To test if the viral inoculum could influence the neuroinvasiveness of SARS-CoV-2 and the incidence of olfactory dysfunctions, we infected hamsters with a 2-log lower infectious dose of Wuhan ($6 \times 10^2$ PFU). These animals were subjected to the same clinical-behavioral tests described above. In the acute phase of the infection, up to 4 dpi, the low infectious dose induced similar body weight loss and clinical score as the animals infected with the initial inoculum ($6 \times 10^4$ PFU), as well as infectious viral titers in the airways and in the olfactory bulbs (Supplementary Fig. 5). Surprisingly, despite the presence of infectious virus in the olfactory bulbs, animals infected with a lower infectious dose presented a lower incidence of olfactory dysfunction (25%, 2/8).

### SARS-CoV-2 travels retrogradely and anterogradely along axons

In vitro, the infection of neurons with SARS-CoV-2 is a contradictory subject[32]. On the one hand, there are descriptions that SARS-CoV-2 can infect hPSC-derived dopaminergic neurons and cause neuronal senescence[33], or that a small subset of iPSC-derived neurons can be subjected to an abortive infection[34]. On the other hand, it was suggested that the presence of additional permissive cells may be necessary for a successful neuronal infection[35]. The attempts we made to infect monocultures of hNSC-derived neurons were not positive, and consequently, we developed a co-culture system with hNSC-derived neurons and SARS-CoV-2-receptive epithelial cells (A549-hACE2-TMPRSS2 cells). We grew these co-cultures using axonal diodes in microfluidic devices[36] to assess the ability of SARS-CoV-2 to infect

## Light sheet imaging at 4 dpi

### A. Nasal turbinates

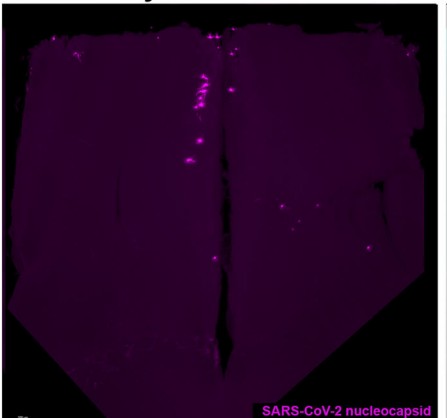

### B. Olfactory bulbs

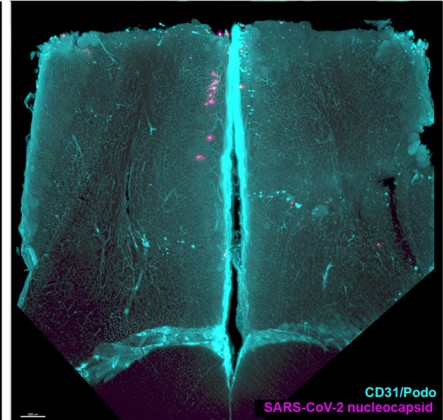

## Bioluminescence imaging of recombinant SARS-CoV-2/nLuc at 4 dpi

### C. Clinical profile

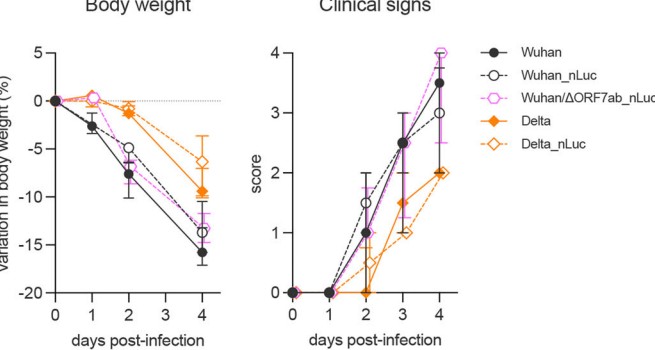

### D. *Ex vivo* bioluminescence

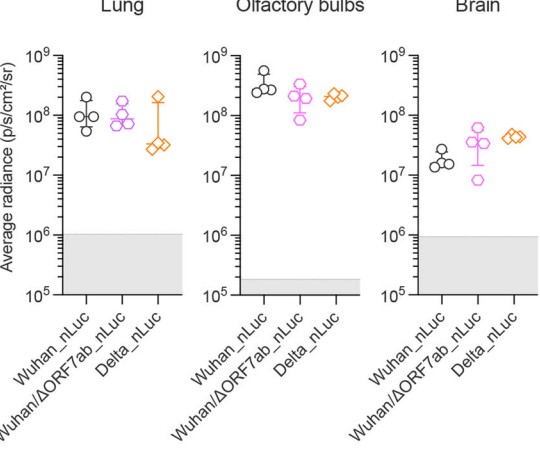

### E. Lung

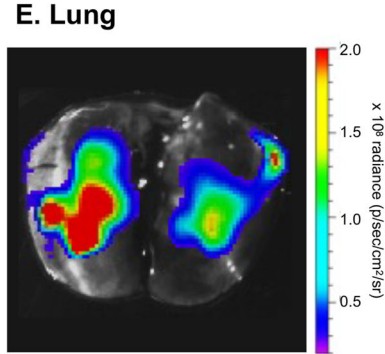

### F. Brain

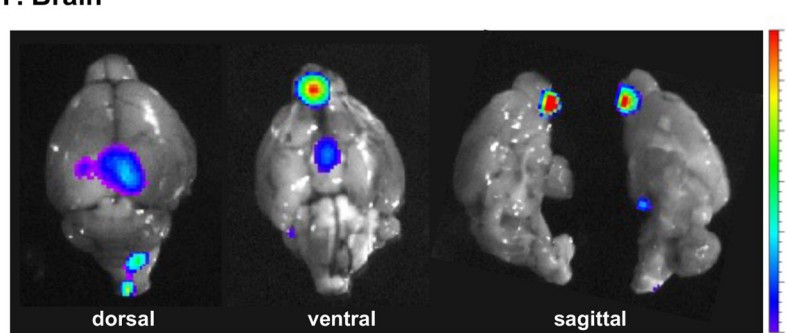

neurons and to move inside axons. In these devices, the co-cultures form neuron-epithelial networks in both the left and the right chambers of the device (Fig. 4A, B), that are connected exclusively via an unidirectional growth of axons from the left towards the right chamber (Fig. 4C). Then, the right or the left chamber of the microfluidic devices was infected with the ancestral virus (Wuhan), the Wuhan/ΔORF7ab and the VoCs Gamma, Delta and Omicron/BA.1. At 3 dpi, infected cells were detected in both the right and the left chambers of all devices, attesting that the virus travelled from the right to the left chamber inside the axons (retrograde axonal transport; Fig. 4D, H and Supplementary Fig. 6), and from the left to the right chamber inside the axons

(anterograde axonal transport; Fig. 4I–M and Supplementary Fig. 7). To further confirm the involvement of the axonal transportation of SARS-CoV-2, we used ciliobrevin D, an inhibitor of the cytoplasmic dynein transport machinery which acts on the bidirectional transport of organelles along axons[37]. When the infection occurred in the presence of ciliobrevin D, the retrograde transport was blocked, and no SARS-CoV-2-infected cell was observed in the left chamber (Supplementary Fig. 8). To quantify the arrival of viral particles via the axons, we infected microfluidic devices in both retrograde and anterograde conditions with the recombinant viruses expressing nanoluciferase Wuhan_nLuc, Wuhan/ΔORF7ab_nLuc, and Delta_nLuc, and we

**Fig. 3 | Imaging assessment of SARS-CoV-2 neuroinvasion. A, B** iDISCO+ whole head clearing and immunolabeling against the SARS-CoV-2 nucleocapsid in hamsters infected with the SARS-CoV-2 original virus (Wuhan) at 4 days post-infection (dpi). **A** Representative nasal turbinate (sagittal section from the skull) showing a diffuse distribution of SARS-CoV-2 (scale bar = 500 μm). **B** Representative olfactory bulb section stained for SARS-CoV-2 nucleocapsid and for podoplanin (CD31/Podo) to identify the vascular compartment. Note presence of SARS-CoV-2 in olfactory bulb neurons, with no macroscopic alterations in the vascular network of the olfactory bulb. Scale bar: 200 μm. See Supplementary Movie 1. **C, F** Bioluminescent recombinant SARS-CoV-2 viruses induce a clinical disease with infection of the olfactory bulbs. **C** Clinical profile of hamsters infected with the recombinant viruses SARS-CoV-2 Wuhan_nLuc and Wuhan/ΔORF7ab_nLuc (based on the original SARS-CoV-2 Wuhan) and Delta_nLuc (based on the Delta variant). The body weight loss

and the clinical score are comparable with those induced by wild-type viruses (n = 4/group). Horizontal lines indicate median and the interquartile range (See Fig. 1A). The clinical score is based on a cumulative 0–4 scale: ruffled fur; slow movements; apathy; and absence of exploration activity (See Fig. 1D). **D** Ex vivo bioluminescence values from the lungs, the olfactory bulbs (ventral view), the brains (ventral view) at 4 dpi (n = 4/group). Horizontal lines indicate median and the interquartile range. Kruskal-Wallis test followed by the Dunn's multiple comparisons test (the adjusted p value is indicated when significant). The gray cross-hatched area corresponds to background signals obtained from the same tissues of a mock-infected hamster. **E, F** Representative ex vivo imaging of a lung (**C**) and a brain (**D**) of a hamster infected with a recombinant SARS-CoV-2 expressing the nLuc at 4 dpi. Note that in the brain, the major bioluminescent focus is localized in the ventral face of olfactory bulbs. See Supplementary Fig. 3.

measured the luminescence in the supernatants on a daily basis (Supplementary Fig. 9). Positive nLuc activity was detected in the supernatants of all receiving chambers from 1 dpi, increasing progressively overtime (Supplementary Fig. 9G, H), indicative of retrograde and anterograde SARS-CoV-2 movement. The luminescence values in the infected chambers were constant and at the maximum limit of detection (Supplementary Fig. 1).

## Discussion

The neurotropism of SARS-CoV-2 is still a matter of debate[32]. While studies on natural infection in humans and experimental infection in animal models reported brain infection by SARS-CoV-2[14,38–42], including in in vitro human models of infection[43–45], others failed to detect SARS-CoV-2 in the nervous tissue[5,46]. This inconsistency may be due to the time of infection, as in most cases the available samples are post-mortal, and we have demonstrated, in the hamster model, that viral isolation is time-dependent, with higher viral loads found in acute time-points[14]. Despite this open question, the impact of SARS-CoV-2 infection on the brain is undeniable[2,47–49]. Most of these published data is related to the original SARS-CoV-2 (Wuhan), however, less information concerning VoCs neuropathogenesis is available[10]. Here we show that all evaluated SARS-CoV-2 viruses (Wuhan, Gamma, Delta and Omicron/BA.1), including novel generated recombinant SARS-CoV-2, are able to cause neuroinflammation following brain invasion. This is most likely via the olfactory bulb; however, the exact nerval route warrants further research.

Despite this shared neuroinvasiveness via the olfactory bulbs, disease profile and airways responses are quite dependent on the SARS-CoV-2 variant. Indeed, we show that all VoCs can infect golden hamsters and promote lung inflammation, including Omicron/BA.1 differently from other reports[19,50], probably due to infectious doses or viral isolates. A "variant-effect" in the clinical presentation and in the tissue-related inflammation was evident, with disease severity presenting the following order: SARS-CoV-2 Wuhan > Gamma > Delta > OmicronBA.1, which supports the hypothesis that Omicron/BA.1 evolution has resulted in a tropism more restricted to the upper respiratory tract, thereby causing a less severe clinical disease[10,51–53]. Although with different severities, neuroinvasiveness, neurotropism and neurovirulence[32] seem to be conserved features among SARS-CoV-2 variants. Indeed, the data presented herein give support to the neuroinvasive ability of SARS-CoV-2, as besides detection of viral RNA and isolation of infectious virus from the olfactory bulbs, we could clearly observe SARS-CoV-2 infected neurons in the brain of hamsters by light sheet microscopy. Nevertheless in hamsters, unlike what has been reported in K18-hACE2 mice which ACE2 expression pattern is non-physiological and ectopic[39], detection of SARS-CoV-2-infected cells was restricted to the olfactory bulbs, without evidence of brain vasculature remodeling nor virus associated with blood vessels.

Olfactory bulb infection is therefore a common feature in the SARS-CoV-2 infectious process, regardless of the variant considered. An inflammation response was also observed in this tissue, with a common upregulation of the antiviral gene *Mx2*, regardless of the VoC. The

reason why some VoCs do not cause olfaction loss is still an open question. The infection and inflammation of the olfactory bulbs, if involved in SARS-CoV-2-associated anosmia, does not seem to be enough to cause olfaction loss in the golden hamster. Aggressions to the olfactory mucosa, rather than the olfactory bulb, are indeed likely the main factors contributing to anosmia, as recovery from anosmia has been related to regeneration of the olfactory epithelium in hamsters[54,55]. Moreover, significant changes in the olfactory epithelium, such as apoptosis, architectural damages, inflammation, downregulation of odorant receptors and functional changes in olfactory sensory neurons, might also contribute, on top of the infection per se, to the occurrence of anosmia[10,14,56–59]. Therefore, it seems that infection and inflammation of the olfactory mucosa combined are needed to trigger anosmia, and the lower incidende of anosmia observed after infection by some VoCs is therefore linked to lower levels of inflammation.

Furthermore, besides mutations in the spike sequence in the SARS-CoV-2 variants genome, additional mutations may be present in other regions of the viral genome, including deletions in the ORF7a and ORF7b sequences[27–30]. ORF7a and ORF7b have been related to viral-induced apoptosis, and to interference with the innate immunity and the antiviral response[22–24,60,61], without, however, being essential to viral infection and replication[62,63]. Further, ORF7b has the potential to interfere with cell adhesion proteins in the olfactory mucosa[25], and interestingly, a binary interaction between ORF7b and the human olfactory receptor OR1D5 has also been reported[26]. Deletions or mutations in these regions may therefore play an additional role in the induction of olfactory disturbances, by preserving the architecture and structures of the olfactory epithelium. Another important point is that we can observe a similar clinical profile and low incidence of olfactory disturbances as SARS-CoV-2 Wuhan/ΔORF7ab by simply reducing the viral inoculum of SARS-CoV-2 Wuhan. Regardless of a comparative evolution of body weight loss and clinical score[64], anosmia was less frequent in animals infected with a low dose of SARS-CoV-2 Wuhan. This might be related to the less severe initial aggression suffered by the olfactory epithelium due to lower infectious doses[55,65] and may also explain why other studies did not detect olfaction dysfunction despite detecting inflammatory changes in the brain[66].

Despite the occurrence or not of anosmia, SARS-CoV-2 can infect olfactory sensory neurons and the olfactory bulb[10,14], and we show herein that the virus can infect neurons and travel inside the axons in both retrograde and anterograde directions. SARS-CoV-2 neuroinvasion was hypothesized to occur by axonal transport via cranial nerves (olfactory, vagus, trigeminal) or by the hematogeneous route[32,67,68]. The data presented herein do not support hematogenous diffusion of SARS-CoV-2 towards the brain, but corroborate the hypothesis of the olfactory pathway as a preferential portal of entry towards the olfactory bulbs[69]. Brain infection via the olfactory pathway therefore seems a common feature of coronaviruses[70,71], regardless of clinical disease presentations. Finally, this study highlights that neuroinvasion and anosmia are therefore independent phenomena upon SARS-CoV-2 infection.

## Neuron-epithelial networks in a microfluidic device

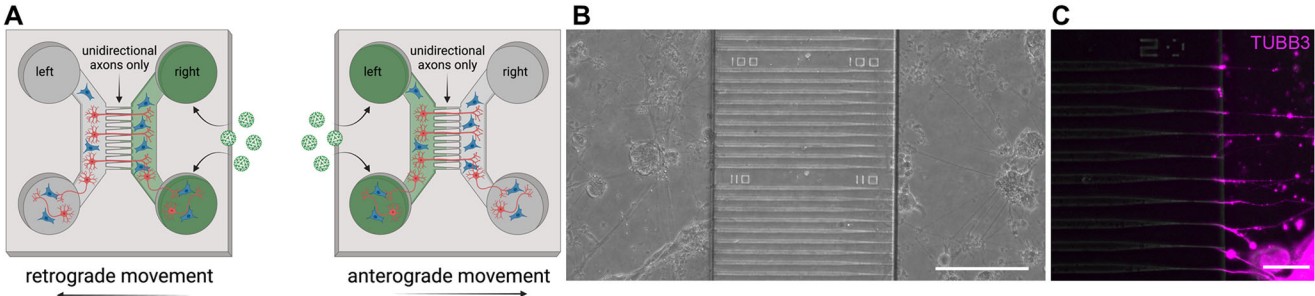

## SARS-CoV-2 retrograde movement ←

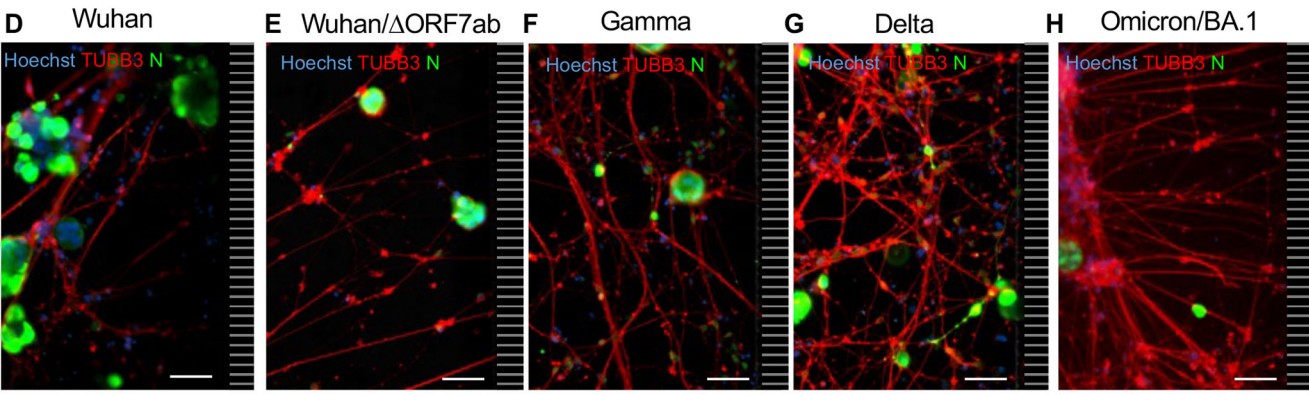

## SARS-CoV-2 anterograde movement →

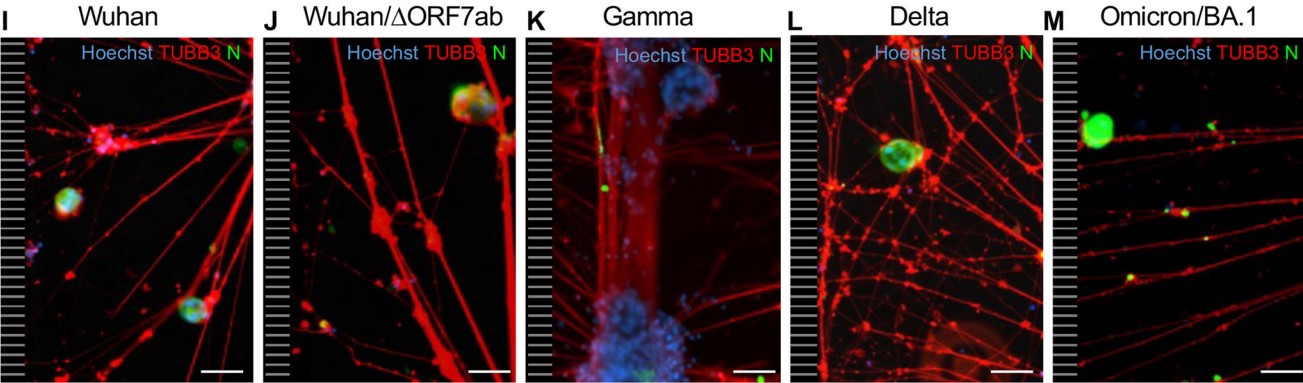

**Fig. 4 | SARS-CoV-2 travels inside axons in in vitro neuron-epithelial networks.** **A** Schematic view of an axon diode in a microfluidic device containing neurons and epithelial cells in both the left and right chambers. The chambers are connected exclusively by the axons of neurons whose cell bodies are located in the left chamber. In this study, SARS-CoV-2 was added in the right chamber (to assess the retrograde axonal transport) or in the left chamber (to assess the anterograde axonal transport). **B** Bright field view of a neuron-epithelial network showing neuronal and A549-ACE2-TMPRSS2 co-culture in both chambers, separated by funnel-shaped micro-channels (scale bar = 200 μm). **C** Detail of β-Tubulin III-positive axons crossing the funnel-shaped micro-channels and entering in the right chamber (scale bar = 50 μm). Neuron-epithelial networks infected at the right chamber with SARS-CoV-2 Wuhan (**D**), Wuhan/ΔORF7ab (**E**), Gamma (**F**), Delta (**G**) and Omicron/BA.1 (**H**) showing infected cells in the left chambers (scale bar = 50 μm). Neuron-epithelial networks infected at the left chamber with SARS-CoV-2 Wuhan (**I**), Wuhan/ΔORF7ab (**J**), Gamma (**K**), Delta (**L**), and Omicron/BA.1 (**M**) showing infected cells in the right chambers (scale bar = 50 μm). Hoechst: nuclei (blue). TUBB3: β-Tubulin III (red). N: SARS-CoV-2 nucleocapsid (green). The black and white striped zones in the right (**D**–**H**) and in the left (**I**–**M**) of the photomicrographs represents the microchannels, which are not accessible during immunostainings. **B**–**M** These photomicrographs are representative of 3 independent experiments. See Supplementary Figs. 6-9. Panel **A** was created with BioRender.com.

## Methods

### Ethics

All animal experiments were performed according to the French legislation and in compliance with the European Communities Council Directives (2010/63/UE, French Law 2013–118, February 6, 2013) and according to the regulations of Institut Pasteur Animal Care Committees. The Animal Experimentation Ethics Committee (CETEA 89) of the Institut Pasteur approved this study (200023; APAFIS#25326-

2020050617114340 v2) before experiments were initiated. Hamsters were housed by groups of 4 animals in isolators and manipulated in class III safety cabinets in the Pasteur Institute animal facilities accredited by the French Ministry of Agriculture for performing experiments on live rodents. All animals were handled in strict accordance with good animal practice.

## SARS-CoV-2 virus and variants

The isolate BetaCoV/France/IDF00372/2020 (EVAg collection, Ref-SKU: 014V-03890) was kindly provided by the National Reference Centre for Respiratory Viruses hosted by Institut Pasteur (Paris, France) and headed by Pr. Sylvie van der Werf. The human sample from which strain hCoV-19/Japan/TY7-501/2021 (Brazilian variant, JPN [P.1]) was supplied by the Japanese National Institute of Infectious Diseases (Tokyo, Japan). The isolate SARS-CoV-2 Delta/2021/I7.2 200 (Indian variant, GISAID ID: EPI_ISL_2029113) and the isolate SARS-CoV-2 Omicron/B.1.1.529 (Omicron BA.1 variant, GISAID ID: EPI_ISL_6794907) were kindly provided by the Virus and Immunity Unit hosted by Institut Pasteur and headed by Olivier Schwartz[72,73]. Viral stocks were produced on Vero-E6 cells infected at a multiplicity of infection of $1 \times 10^{-4}$ PFU/cell. The virus was harvested 3 days post-infection, clarified and then aliquoted before storage at −80 °C. Viral stocks were titrated on Vero-E6 cells by classical plaque assays using semisolid overlays (Avicel, RC581-NFDR080I, DuPont)[74].

Recombinant viruses expressing Nanoluciferase (nLuc) as a reporter gene were obtained by reverse genetics. Wuhan_nLuc was derived from the clone synSARS-CoV-2-GFP-P2A-ORF7a (clone #41) generated as described[75–79], where the GFP reporter gene was replaced by the nLuc one. The cloning strategy was optimized to build a complete cDNA of the Delta_nLuc virus on the same backbone than the isolate EPI_ISL_2029113. SARS-CoV-2 Wuhan/ΔORF7ab was obtained using the same reverse genetic system by replacing the complete ORF7ab sequence by that of GFP or by nLuc (Wuhan/ΔORF7ab_nLuc).

## Generation of the complete genome of SARS- CoV-2/Delta_n-Luc virus

The viral genome was divided in 11 overlapping fragments around 3 kb each. Those fragments and two fragments coding for different yeast-specific selection genes (His3 and Leu2) were recombined in the yeast centromere plasmid pRS416 to generate the complete genome clone of recombinant virus. The T7 promotor was placed just before the 5'UTR sequence and a unique restriction site EAG1 was placed just after the poly(A) tail.

Viral fragments were obtained either by RT-PCR on RNA extracted from Vero-E6 cells infected by Delta using the SuperScript IV VILO Master Mix (11756050, Thermofisher) according to the manufacturer protocol, or using synthetic genes (GeneArt, Life Technologies). All PCR amplifications were performed using Phusion™ High-Fidelity DNA Polymerase (F530, Thermofisher). The PCR product were cloned in Topo-TA vector and their sequences controlled by sanger sequencing. The nLuc gene was inserted upstream ORF7a and separated from it using a GSG linker and a P2A peptide sequence to mediate cleavage between the reporter and the viral protein. It was directly synthetized in the F10-1 fragment.

## Yeast recombination

Recombination of the 4 yeast specific fragments and the 11 viral fragments was performed on *S. cerevisiae* BY4741. A yeast culture was carried out from an overnight pre-culture in YPDA medium (2 mL of pre-culture in 50 mL of YPDA medium heated to 30 °C) until the exponential growth phase was reached, with an optical density corresponding to approximately $10^8$ cells/mL. The yeasts were harvested by centrifugation of 50 mL of culture (3,000 × *g* at 4 °C during 5 min) before being washed a first time in sterile solution of Tris-EDTA and lithium Acetate TE/LiAc (Tris-HCl pH 7.5 10 mM; EDTA

pH 7.5 0.1 mM; LiAc 100 mM). Cells were resuspended in 500 μL of TE/LiAc solution ($2.10^9$ cells/mL). The yeast suspension was then incubated at 30 °C without stirring during 30 min. During this time, a mix of all DNA fragments in an equimolar proportion was prepared. For each recombination condition, 50 μL of competent cells ($10^8$ cells) were resuspended in a final volume of 300 μL TE/LiAc/PEG solution (100 mM Tris-HCl pH 8; 100 mM Lithium Acetate; 40% v/v PEG 3350) with the DNA mix to be recombined and 50 μg of denatured salmon sperm DNA (31149, Sigma) and then cooled on ice for 5 min. The mixture was incubated at 30 °C during 30 min without agitation before undergoing a heat shock at 42 °C during 20 min. After cooling on ice, the yeasts were pelleted by centrifugation at 500 × *g* at RT during 5 min and then resuspended and incubated with 500 μL of 5 mM CaCl₂ at RT for 10 min. CaCl₂ was washed by centrifugation at 500 × *g* at RT during 5 min and cells were resuspended in 200 μL of sterile water before being plated on selective synthetic minimal medium lacking histidine, Uracyl and Leucine (SD-His-Ura-Leu-), and incubated at 30 °C for 3 days. For each transformation, ten different clones were checked by multiplex PCR (206143, Qiagen) on rapid DNA preparations carried out by the Chellex technique as already described[75] using specific primers (Supplementary Table 1) to control the presence of the different fragments in the expected orientation. The yeast colony of interest was sub-cultured in 200 mL of SD-His-Ura-Leu- medium and the plasmid extracted using the Qiagen midi prep kit (12143, Qiagen) following manufacturer's instructions.

## Rescue of the virus

The plasmid was linearized by EagI-HF (R3505, NEB) digestion, purified by a phenol chloroform process and transcribed in vitro into RNA using the T7 RiboMaxTM Large Scale RNA Production System (P1300, Promega) and the m7G(5')ppp(5')G RNA Cap Structure Analog (#S1411, NEB) kits. Approximately 5 μg of linear DNA were used as template in a reaction volume of 50 μL comprising: 10 μL of 5X T7 transcription buffer; 5 μL of m7G(50)ppp(50)G RNA cap analog structure at 30 mM; 0.75 μL of 100 mM GTP; 3.75 μL of each nucleotide type ATP, CTP; 100 mM UTP; and 5 μL of RNasin T7 RNA polymerase enzyme. The reaction was incubated at 30 °C during 3 h then the template DNA was digested at 37 °C during 20 min by adding 2 μL of the enzyme RQ1 RNase free DNase. The synthesized RNA was finally purified using a classical phenol chloroform method and precipitated. Twelve μg of complete viral mRNA and 4 μg of plasmid encoding the viral nucleoprotein (N) gene were electroporated on $8 \times 10^6$ Vero-E6 cells resuspended in 0.8 mL of Mirus Bio™ Ingenio™ Electroporation Solution (MIR50114) using the Gene Pulser Xcell Electroporation System (1652660, Biorad) with a pulse of 270 V and 950 μF. The cells were then transferred to a T75 culture flask with 15 mL of DMEM supplemented with 2% FCS (v/v). The electroporated cells were incubated at 37 °C, 5% CO₂ for several days until the cytopathic effect (CPE) was observed. Then, the supernatant was harvested, aliquoted and frozen at −80 °C until titration. The viral sequence was controlled by NGS.

## Viral growth curves

Vero-E6 cells (ATCC CRL-1586) were used to assess the replication kinetics of different SARS-CoV-2 variants and recombinant viruses. The cells were maintained in Dulbecco's modified culture medium (DMEM, 31966-021, Gibco) supplemented with 5% fetal calf serum, at 37 °C in 5% CO₂. On the day before infection, $1 \times 10^6$ VeroE6 cells/well were plated onto 6-well plates. On day of infection, the cells were infected with SARS-CoV-2 virus at a multiplicity of infection (MOI) of 0.01, in independent triplicates, and incubated during 24-, 48- or 72-hours post-infection. The supernatants were collected at each time-point and frozen at −80 °C. Viral titers were obtained by classical TCID₅₀ method on Vero-E6 cells after 72 hours post-infection[80].

## SARS-CoV-2 model in golden Syrian hamsters

Male golden Syrian hamsters (*Mesocricetus auratus;* RjHan:AURA) of 5–6 weeks of age (average weight 60–80 grams) were purchased from Janvier Laboratories and handled under specific pathogen-free conditions. The animals were housed and manipulated in isolators in a biosafety level-3 facility, with *ad libitum* access to water and food. Before any manipulation, animals underwent an acclimation period of one week. Animals were anesthetized with an intraperitoneal injection of 200 mg/kg ketamine (Imalgène 1000, Merial) and 10 mg/kg xylazine (Rompun, Bayer), and 100 µL of physiological solution containing $6 \times 10^4$ PFU of SARS-CoV-2 (wild-type or recombinant) was administered intranasally to each animal (50 µL/nostril). Mock-infected animals received the physiological solution only. Infected and mock-infected hamsters were housed in separate isolators and were followed-up daily during four days at which the body weight and the clinical score were noted. The clinical score was based on a cumulative 0-4 scale: ruffled fur, slow movements, apathy, absence of exploration activity. At day 3 post-infection (dpi), animals underwent a food finding test to assess olfaction as previously described[14,81]. Briefly, 24 hours before testing, hamsters were fasted and then individually placed into a fresh cage ($37 \times 29 \times 18$ cm) with clean standard bedding for 10 minutes. Subsequently, hamsters were placed in another similar cage for 2 minutes when about 5 pieces of cereals were hidden in 1.5 cm bedding in a corner of the test cage. The tested hamsters were then placed in the opposite corner and the latency to find the food (defined as the time to locate cereals and start digging) was recorded using a chronometer. The test was carried out during a 15 min period. As soon as food was uncovered, hamsters were removed from the cage. One minute later, hamsters performed the same test but with visible chocolate cereals, positioned upon the bedding. The tests were realized in isolators in a Biosafety level-3 facility that were specially equipped for that. At 4 dpi, animals were euthanized with an excess of anesthetics (ketamine and xylazine) and exsanguination[82], and samples of nasal turbinates, lungs and olfactory bulbs were collected and immediately frozen at −80 °C. Fragments of lungs were also collected and fixed in 10% neutral-buffered formalin.

## SARS-CoV-2 detection in golden hamsters' tissues

Frozen lung fragments, nasal turbinates and olfactory bulbs were weighted and homogenized with 1 mL of ice-cold DMEM supplemented with 1% penicillin/streptomycin (15140148, Thermo Fisher) in Lysing Matrix M 2 mL tubes (116923050-CF, MP Biomedicals) using the FastPrep-24™ system (MP Biomedicals), and the following scheme: homogenization at 4.0 m/s during 20 sec, incubation at 4 °C during 2 min, and new homogenization at 4.0 m/s during 20 sec. The tubes were centrifuged at $10,000 \times g$ during 2 min at 4 °C, and the supernatants collected. Viral titers were obtained by classical $TCID_{50}$ method on Vero-E6 cells after 72 hours post-infection[80]. Viral RNA loads were obtained by the quantification of genomic and subgenomic SARS-CoV-2 RNA based on the E gene[83]. Briefly, 125 µL of the supernatants were homogenized with 375 µL of Trizol LS (10296028, Invitrogen) and total RNA was extracted using the Direct-zol RNA MicroPrep Kit (R2062, Zymo Research: nasal turbinates and olfactory bulbs) or MiniPrep Kit (R2052, Zymo Research: lung). We used the Taqman one-step qRT-PCR (Invitrogen 11732-020) in a final volume of 12.5 µL per reaction in 384-wells PCR plates using a thermocycler (QuantStudio 6 Flex, Applied Biosystems). Briefly, 2.5 µL of RNA were added to 10 µL of a master mix containing 6.25 µL of 2X reaction mix, 0.2 µL of $MgSO_4$ (50 mM), 0.5 µL of Superscript III RT/Platinum Taq Mix (2 UI/µL) and 3.05 µL of nuclease-free water containing 400 nM of primers and 200 nM of probe. To detect the genomic RNA, we used the E_sarbeco primers and probe (E_Sarbeco_F1 5'-ACAGGTACGTTA ATAGTTAATAGCGT-3'; E_Sarbeco_R2 5'-ATATTGCAGCAGTACGCACA CA-3'; E_Sarbeco_Probe FAM-5'-ACACTAGCCATCCTTACTGCGCTTC G-3'-TAMRA). The detection of sub-genomic SARS-CoV-2 RNA was

achieved by replacing the E_Sarbeco_F1 primer by the CoV2sgLead primer (CoV2sgLead-Fw 5'-CGATCTCTTGTAGATCGTTCTC-3'). A synthetic gene encoding the PCR target sequences was ordered from Thermo Fisher Scientific. A PCR product was amplified using Phusion™ High-Fidelity DNA Polymerase (Thermo Fisher Scientific) and in vitro transcribed by means of Ribomax T7 kit (Promega). RNA was quantified using Qubit RNA HS Assay kit (Thermo Fisher scientific), normalized, and used as a standard to quantify RNA absolute copy number. The amplification conditions were as follows: 55 °C for 20 min, 95 °C for 3 minutes, 50 cycles of 95 °C for 15 s and 58 °C for 30 s; followed by 40 °C for 30 s.

## Transcriptomics analysis in golden hamsters' tissues

RNA preparations from lungs, nasal turbinates and olfactory bulbs collected at 4 dpi were submitted to RT-qPCR. Briefly, RNA was reverse transcribed to first strand cDNA using the SuperScript™ IV VILO™ Master Mix (11766050, Invitrogen). qPCR was performed in a final volume of 10 µL per reaction in 384-well PCR plates using a thermocycler (QuantStudio 6 Flex, Applied Biosystems) and its related software (QuantStudio Real-time PCR System, v.1.2, Applied Biosystems). Briefly, 2.5 µL of cDNA (12.5 ng) were added to 5 µL of Taqman Fast Advanced master mix (444457, Applied Biosystems) and 2.5 µL of nuclease-free water containing golden hamster's primer pairs (Supplementary Table 2). The amplification conditions were as follows: 95 °C for 20 s, and 45 cycles of 95 °C for 1 s and 60 °C for 20 s. The *γ-actin* and the *Hprt* (hypoxanthine phosphoribosyl-transferase) genes were used as reference. Variations in gene expression were calculated as the n-fold change in expression in the tissues from the infected hamsters compared with the tissues of the mock-infected group using the $2^{-\Delta\Delta Ct}$ method[84].

## Sample staining and iDISCO+ clearing

For iDISCO +, three male and three female hamsters were infected with $6 \times 10^4$ PFU of SARS-CoV-2 Wuhan and followed up as described above. At 4 dpi, animals were anesthetized with an intraperitoneal injection of ketamine (200 mg/kg; Imalgène 1000, Merial) and xylazine (10 mg/kg; Rompun, Bayer), and we performed a transcardial perfusion with DPBS containing heparin ($5 \times 10^3$ U/mL) followed by 4% neutral-buffered formaldehyde. The whole heads were collected and stored in 4% neutral-buffered formaldehyde during one week before analysis. SARS-CoV-2 nucleocapsid was detected *in toto* in the snout and brain of infected hamsters using the iDISCO+ protocol previously published[85] with minimal modifications. All buffers were supplemented with 0,01% of sodium azide (Sigma-Aldrich). The snout, containing the olfactory mucosa and the olfactory nerve, and the brain were dissected and processed separately to optimize tissue manipulation.

All samples were first dehydrated in methanol (Sigma-Aldrich) using 20-40-60-80-100-100% dilutions in distilled water (1 h to 2 h each concentration). Complex lipid removal was achieved incubating the samples in a 2:1 mixture of dichloromethane (Sigma-Aldrich) and methanol overnight. Samples were then washed twice in methanol 100% and bleached overnight using a $H_2O_2$ 5% solution in methanol. Then all samples were rehydrated using a methanol series (80-60-40-20%). To increase bone permeability, the snouts were decalcified by incubation in the Morse's solution (formic acid 45% and sodium citrate 20%). Finally, all the samples (brains and snouts) were washed in PBS, then PBS-T (PBS with 0,2% Triton X-100 [Sigma-Aldrich]) and incubated in permeabilization buffer (20% DMSO [Sigma-Aldrich] and 2.3% glycine [Sigma-Aldrich] in PBS-T) at 37 °C overnight. Before immunostaining, samples were blocked (0,2% Gelatin [Sigma-Aldrich] in PBS-T) at 37 °C for 24 h. To reveal SARS-CoV-2 and vasculature, three primary antibodies were combined: Rabbit anti-SARS CoV2 nucleocapsid antibody (GTX135357, GeneTex) diluted 1:1000; Goat anti-CD31 (AF3628, R&D Systems) diluted 1:300; and Rat anti-Podocalyxin

(MAB1556, R&D Systems) diluted 1:1000. After a 2-weeks incubation in primary antibody, the samples were washed (PBS supplemented with 0,2% tween-20 [Sigma-Aldrich] and 2 U/mL heparin [Sigma-Aldrich]) and incubated 10 days with the following secondary antibodies: Donkey anti-Rabbit Alexa 555 (A-31572, Thermo Fisher Scientific), Donkey anti-Goat Alexa 647 (A-21447, Thermo Fisher Scientific) and Chicken anti-Rat Alexa 647 (A-21472, Thermo Fisher Scientific) all diluted at 1:500. All antibodies were diluted in blocking solution and incubated at 37 °C with gentle shacking. After immunostaining, the samples were washed, dehydrated in methanol (20-40-60-80-100-100%), incubated for 3 h in a 2:1 mixture of dichloromethane and methanol, washed twice (15 min each) in dichloromethane 100% and cleared by immersion in dibenzyl ether.

## Light sheet imaging
Brain and snout samples were imaged using a LaVision Ultramicroscope II equipped with infinity-corrected objectives, laser lines OBIS-561nm 100 mW and OBIS-639nm 70 mW, and 595/40 and 680/30 filters for Alexa 555 and Alexa 647 respectively. The olfactory bulbs were imaged with a 4×0,35NA objective, using a laser NA of 0,3 and a step size of 2 μm obtaining images with $1,63 \times 1,63 \times 2$ μm/pixel resolution. The snouts were imaged with a 1,3X objective, adjusting the laser NA to 0,3 and step size of 5 μm obtaining images with a $5 \times 5 \times 5$ μm/pixel resolution. All acquisitions were done with ImSpector software (v.7.0.127.0, Lavision Biotec GmbH). The 3D stacks obtained were analyzed using Bitplane Imaris 9.2 (Oxford instruments).

## In vivo and ex vivo bioluminescence imaging
At 4 dpi, animals infected with SARS-CoV-2/Wuhan_nLuc, Wuhan/ΔORF7ab_nLuc and SARS-CoV-2/Delta_nLuc were anesthetized with an intraperitoneal injection of ketamine (200 mg/kg; Imalgène 1000, Merial) and xylazine (10 mg/kg; Rompun, Bayer). Next, 0.45 μmoles of the nLuc substrate (Nano-Glo® in vivo substrate, CS320501, Promega) fluorofurimazine (FFz) were injected interiperitoneally and the animals were placed in a confinement box and imaged using an IVIS® Spectrum In Vivo Imaging System (PerkinElmer) within 5 minutes of FFz injection. Two-dimensional bioluminescence images were recorded, and photon emission was quantified (p/s/cm²/sr) in a region of interest defined using Living Image software (v. 4.7.4, PerkinElmer). After in vivo imaging, animals were euthanized, the lungs and the brains were collected and quickly placed in a six-wells plate. The plates were placed in a confinement box and imaged as described above.

## Histopathology and Immunohistochemistry
Lung fragments fixed 7 days in 10% neutral-buffered formalin were embedded in paraffin. Four-μm-thick sections were cut and stained with hematoxylin and eosin staining. IHC were performed on Leica Bond RX using anti SARS Nucleocapsid Protein antibody (1:500, NB100-56576, Novus Biologicals) and biotinylated goat anti-rabbit Ig secondary antibody (1:600, E0432, Dako, Agilent). Slides were then scanned using Axioscan Z1 slide scanner (Zeiss) and images were analyzed with the Zen 2.6 software (Zeiss).

## Neuron-epithelial networks in microfluidic chambers
Human neural stem cells (hNSC, ENStem-A, SCC003, EMD-Millipore) were maintained in geltrex-coated 6-well plates (A1413302, Gibco) in a density of $4.5 \times 10^5$ cells/cm² in CTS Knock-out DMEM/F12 medium (A1370801, Gibco) supplemented with 2% StemPro neural supplement (A1050801, Gibco), 2 mM glutamax (35050038, Gibco), 20 ng/mL FGFb (PHG0026, Gibco) and 20 ng/mL EGF (PHG0311, Gibco) at 37 °C in 5% CO₂. A549-ACE2-TMPRSS2 cells (kindly provided by Pr. Olivier Schwartz, Institut Pasteur) were maintained in F12K Nut Mix medium (21127022, Gibco) supplemented with 10% bovine calf serum and 10 μg/mL blasticidine (12172530, Gibco) at 37 °C in 5% CO₂. To generate the neuron-epithelial networks, we used microfluidic chips purchased

from MicroBrain Biotech (Brainies™, Cat#: MBBT5; Marly le Roi, France). Brainies™ MBBT5 is a chip containing 4 neuronal diodes. One neuronal diode includes 2 rectangular culture chambers (volume ~1 μL) each connected to 2 reservoirs and separated by a series of 500 μm-long asymmetrical micro-channels (3 μm high, tapering from 15 μm to 3 μm)[36]. Brainies™ were coated with poly-L-ornithine (P4957, Sigma-Aldrich) and laminin (L2020, Sigma-Aldrich) and $1 \times 10^5$ hNSCs were seeded in both the right and the left chambers and incubated at 37 °C in 5% CO₂ for 24 hours. The medium was then replaced by a neuronal differentiation medium composed of neurobasal medium (10888022, Gibco) supplemented with 2% B27 supplement (17504044, Gibco), 1% CultureOne supplement (A3320201, Gibco), 2 mM glutamax (35050038, Gibco), and 200 μM ascorbic acid (A4403, Sigma Aldrich) and incubated at 37 °C in 5% CO₂ for 14 days, replacing half of the medium every other day. Finally, $2 \times 10^4$ A549-ACE2-TMPRSS2 cells were added over the neurons in both the right and the left chambers in neuronal differentiation medium at 37 °C in 5% CO₂ for 48 hours. To assess the SARS-CoV-2 retrograde axonal movement, the right chambers were infected at a MOI of 1 of SARS-CoV-2 Wuhan, Wuhan/ΔOR-F7ab, Gamma, Delta, or Omicron/BA.1, and incubated at 37 °C in 5% CO₂ for 72 hours. To assess the SARS-CoV-2 anterograde axonal movement, the left chambers were infected in the same above-mentioned conditions. To prevent passive diffusion of assemblies, hydrostatic pressure was applied by adding medium excess in the reservoirs of recipient chambers[86]. To block the axonal retrograde transport, 100 μM of ciliobrevin D (250401, Calbiochem) was added in the medium during infection and throughout the incubation period. The cells were then fixed with 4% PFA (15444459, Thermo Scientific), washed in PBS, permeabilized with 0.5% Triton X-100 during 10 minutes, washed once in PBS, blocked with 10% normal goat serum (10000 C, Invitrogen) during 30 minutes followed by overnight incubation at 4 °C with primary antibodies: mouse anti-β-Tubulin III (neuronal) antibody (1:1000, T8578, Sigma-Aldrich) and rabbit anti-SARS-CoV-2 nucleocapsid antibody (1:1000, GTX135361, GeneTex). The cells were then washed in PBS and incubated with the following secondary antibodies: goat anti-mouse AlexaFluor 647 (1:1000, A21235, Invitrogen) and goat anti-rabbit AlexaFluor 546 (1:1000, A11035, Invitrogen) for 2 hours at 4 °C. The cells were then washed, the nuclei were stained with 20 μM Hoechst 33342 (62249, Thermo Scientific) and stored in PBS. Images were obtained using the EVOS FL cell imaging system (Thermo Scientific) with the objectives 10x or 20x and the fluorescence cubes DAPI, RFP and CY5.5. Using the same described method, neuron-epithelial networks in microfluidics devices were also infected with the recombinant viruses Wuhan_nLuc, Wuhan/ΔORF7ab_nLuc, and Delta_nLuc at a MOI of 0.5 (maximum volume limitation due to stock viral titer). The nanoluciferase activity in the supernatants was evaluated sequentially at 1, 2 and 3 dpi. Briefly, 20 μL of supernatant was mixed with 20 μL of nLuc substrate (N1110, Promega) in a white 96-wells plate and the luminescence, expressed as Relative light Unit (RLU), was acquired during 100 ms using the VICTOR Nivo Plate reader, and the Victor Nivo control software (v.4.0.7, Perkin Elmer).

## Statistics and reproducibility
Statistical analysis was performed using Prism 9 (GraphPad, version 9.5.1, San Diego, USA), with $p < 0.05$ considered significant. Quantitative data was compared across groups using Log-rank test, two-tailed Mann-Whitney test or Kruskal-Wallis test followed by the Dunn's multiple comparisons test. Multivariate statistical analyses on clinical parameters and on tissue inflammation were achieved using Principal Component Analysis. Randomization and blinding were not possible due to pre-defined housing conditions (separated isolators between infected and non-infected animals). Ex vivo and in vitro analyses were blinded (coded samples). For in vivo experiments, 2 independent replicates with 4 animals each were performed; for in vitro experiments, 3 independent replicates were performed.

**Reporting summary**

Further information on research design is available in the Nature Portfolio Reporting Summary linked to this article.

## Data availability

All relevant data supporting the key findings of this study are available within the main manuscript and its Supplementary Information. Source data are provided with this paper.

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

## Acknowledgements

The SARS-CoV-2 strain was supplied by the National Reference Centre for Respiratory Viruses hosted by Institut Pasteur (Paris, France) and

headed by Pr. Sylvie van der Werf. The human sample from which strain 2019-nCoV/IDF0372/2020 was isolated has been provided by Pr. X. Lescure and Pr. Y. Yazdanpanah from the Bichat Hospital (Paris, France). The human sample from which strain hCoV-19/Japan/TY7-501/2021 (Gamma variant, JPN [P.1]) was supplied by the Japanese National Institute of Infectious Diseases (Tokyo, Japan). The isolate SARS-CoV-2 Delta/2021/I7.2 200 (Delta variant, GISAID ID: EPI_ISL_2029113), the isolate SARS-CoV-2 Omicron/B.1.1.529 (Omicron BA.1 variant, GISAID ID: EPI_ISL_6794907) and the A549-ACE2-TMPRSS2 cells were supplied by the Virus and Immunity Unit hosted by Institut Pasteur and headed by Pr. Olivier Schwartz. We thank Sanjay Vashee from the J. Craig Venter Institute (Rockville, MD, USA) for the YCpBAC-his3 plasmid. This work was supported by Institut Pasteur's Task Force SARS-CoV-2 (NeuroCovid Project), by SARS-CoV-2 joint call Institut Pasteur - Paris Brain Institute (CoVessel Project), by Institut Pasteur's Programme Fédérateur de Recherche 1 (PFR-1 - Reverse Genetics). G.D.M. acknowledges funding from the Fondation pour la Recherche Médicale (grant ANRS MIE202112015304). V.P. is recipient of a fellowship from the European Union's Horizon 2020 Framework Programme for Research and Innovation under Specific Grant Agreement No. 945539 (Human Brain Project SGA3). F.A. is recipient of a fellowship from Institut Pasteur's Programme Fédérateur de Recherche 5 (PFR-5 - Functional Genomics of the Viral Cycle). A.C. acknowledges funding from the Institut Pasteur's 2022-2023 Brain Axis SRA3 M2 Master Student Call. E.S.L and R.K acknowledge support from the Institut Pasteur's Task Force (project SABSOS). E.S.L acknowledges funding from the INCEPTION programme (Investissements d'Avenir grant ANR-16-CONV-0005). We would like to acknowledge Yves Jacob for the fruitful discussions about SARS-CoV-2 ORF7. We thank Etienne Jacotot for his insights on neuronal cultures in microfluidic chambers, as well as Bernadette Bung from MicroBrain Biotech. We also thank Johan Bedel for the help with histopathology, and Emeline Perthame for her insights on statistical analyses. Part of this work was performed at the UtechS Photonic BioImaging (PBI) platform supported by Institut Pasteur and by Région Ile-de-France (program DIM1Health).

## Author contributions

Conceptualization: G.D.M., F.L., and H.B. Methodology: G.D.M., R.K., V.T., N.R., and F.L. Investigation: G.D.M., V.P., F.A., A.V.P., S.K., L.K., A.C., M.T., A.P., and A.T. Funding Acquisition: G.D.M., M.L., P.M.L., N.R., F.L., and H.B. Resources: B.S.T., D.H., N.W., S.M., R.K., E.S.L., N.R., V.T., and F.L. Supervision: H.B. Writing—Original Draft: G.D.M., and H.B. Writing—Review & Editing: all authors.

## Competing interests

The authors declare no competing interests.

## Additional information

[1]Institut Pasteur, Université Paris Cité, Lyssavirus Epidemiology and Neuropathology Unit, F-75015 Paris, France. [2]Institut Pasteur, Université Paris Cité, Channel Receptors Unit, F-75015 Paris, France. [3]Sorbonne Université, Collège Doctoral, F-75005 Paris, France. [4]Institut du Cerveau et de la Moelle Épinière, Laboratoire de Plasticité Structurale, , Sorbonne Université, INSERM U1127, CNRS UMR7225, 75013 Paris, France. [5]Institute of Virology and Immunology (IVI), Bern, Switzerland; Department of Infectious Diseases and Pathobiology, Vetsuisse Faculty, University of Bern, Bern, Switzerland. [6]Institut Pasteur, Université Paris Cité, Histopathology Platform, F-75015 Paris, France. [7]Institut Pasteur, Université Paris Cité, Spatial Regulation of Genomes Laboratory, F-75015 Paris, France. [8]Institut Pasteur, Université Paris Cité, Molecular Genetics of RNA viruses Unit, F-75015 Paris, France. [9]Institut Pasteur, Université Paris Cité, Evolutionary Genomics of RNA Viruses Group, F-75015 Paris, France. [10]Multidisciplinary Center for Infectious Diseases, University of Bern, Bern, Switzerland. [11]Institut Pasteur, Université Paris Cité, Inserm U1117, Biology of Infection Unit, 75015 Paris, France. [12]Necker-Enfants Malades University Hospital, Division of Infectious Diseases and Tropical Medicine, APHP, Institut Imagine 75006 Paris, France. [13]Institut Pasteur, Université Paris Cité, Perception and Memory Unit, F-75015 Paris, France; CNRS UMR3571, 75015 Paris, France. [14]These authors contributed equally: Victoire Perraud, Flavio Alvarez, Alba Vieites-Prado. [15]These authors jointly supervised this work: Florence Larrous, Hervé Bourhy. ✉e-mail: herve.bourhy@pasteur.fr

