## [Peer Review File · Nature Communications]

Neuroinvasion and anosmia are independent phenomena upon infection with SARS-CoV-2 and its variantsREVIEWER COMMENTS

Reviewer #1 (Remarks to the Author):

The presented study lead by Larrous and Bourhy focus on neuroinvasion and inflammatory response of the olfactory bulb and nasal mucosa in hamsters infected with different SARS-CoV-2 strains.

The authors characterized clinically the disease course, measured TCID50, viral RNA (including subgenomic RNA) in the specific regions and show that despite different clinical signs (including olfaction) and lung pathology in the different strains, the olfactory bulb showed in all genomic viral RNA. And virus detection via nanoluciferase SARS-CoV2 signal in animals with Delta or Wuhan strains infected. This is interpreted as infection of the olfactory bulb via the epithelium as the authors demonstrated that SARS-CoV-2 travels retrogradely along axons in microfluidic chambers in vitro.

Overall the study is complementary to previous data focusing of olfactory epithelium (Chen et al. biorxiv 2022) or inflammatory brain response (Frere et al. 2022). The experimental design is solid, however I have some concerns about interpretation of the data and technical questions:

1.) Data interpretation: The authors show convincingly that SARS-CoV-2 can be transported via axons in vitro, and that virus can be found in the olfactory bulb in infected animals. However, the authors do not show neuroinflammatory reactions that one would expect during viral CNS infection as microglial nodules/activated microglial, infiltrating T cells, dead neurons as in neurotropic virus. So, if SARS-CoV-2 is “infecting” neurons, this is probably non-reproductive, and not leading to classical neuroinflammation nor functional alterations as animals infected with Omicron/Delta do not show altered olfaction despite similar viral RNA detection in the tissue. The question is if the detected RNA/ luciferase signal is at all a sign of infection. Is SARS-CoV-2 on this model neurotropic? Furthermore, “No virus was observed in other areas of the brain”. But this data is not shown in the manuscript, except for luciferase dataset. But is there RNA detectable? Protein? Is the inflammatory change the authors describe also detectable in areas without viral RNA/protein? This would point to a more general bystander reaction instead of a direct virus mediated effect. And to address the neuroinflammation: Are there signs of neuroinflammation in the olfactory bulb, this is not clear after measuring IFN β , MX2 etc.? Cellular changes should be analyzed as reactivated microglia, infiltrating Tcells etc.

2.) Technical issues:

- the reader should be enabled to connect clinical picture and molecular/imaging data better, therefore animals could e.g. be identifiable with a code and this information should be given in the respective figure legends together with a table with clinical data values. E.g. one would like to check as a reader if the animals detected with luciferase signal in the olfactory bulb had also impaired olfaction or not. And similar viral protein in the bulbus is

only shown for Wuhan strain via light sheet of one animal?

-As the number of animals is in some of the experiments very limited (probably due to animal protection reasons) I recommend that the authors show all data (all animals analyzed) and not only "representative" images especially in the lightsheet dataset.

- What is the rationale of the genes selected for inflammatory reaction?

-the method section is lacking important information regarding fluorescent microscopy of the in vitro part; an excellent guideline regarding this topic is given in Llopis et al. Nature methods 2021.

Minor points:

-in the discussion the authors list studies that supposedly show brain infection with SARS-CoV-2. These studies show indeed viral RNA in different brain regions but no sgRNA and no viral particles and the spike staining is not convincing due to high "background" staining also in controls as reported e.g., by Yang et al. Nature 2021...as viral RNA is not equal brain infection, brain invasion of SARS-CoV-2 in humans has not been shown so far up to my knowledge.

-for the given clone of nucleocapsid the company doesn't show a tissue staining, how did the authors validated/verified antibody specificity and sensitivity?

-it is not clear for me, if only male hamsters were used as written in the reporting summary or also female as written in the methods section, here again an overview with all included animals with respective analysis would help. Were the female hamsters only used for light-sheet analysis?

- video S2 needs more explanation please, what is the red structure?

-How was PCA computation done? How were PCA plots generated?

-Sometimes the authors use nucleocapsid (correct) but sometimes also nucleoprotein, this should be adapted.

-Single channel images for all shown fluorescent images should be made available in the supplement or via online depository.

Reviewer #2 (Remarks to the Author):

De manuscript by Dias de Melo et al. is an elegant study which combines different techniques to monitor SARS-CoV-2 infection in hamsters, and to visualize virus invasion into the CNS. This study reveals that all SARS-CoV-2 variants can invade the CNS via the olfactory nerve, independent of their capacity to cause disease, including anosmia. However, due to the limited numbers of animals, it is difficult to determine whether there are differences among the SARS-CoV-2 variants included in the studies. In addition, I have some concerns whether the assessment of olfaction includes all necessary controls.

Specific comments:

Introduction: It is not introduced why the authors included SARS-CoV-2 with a ORF7ab deletion in their studies.

Line 104 (and 322): The olfactory nerve is different from the other cranial nerves. The cell body of the olfactory sensory neuron resides in the nasal mucosa and the axon penetrates the cribriform plate where it has contact with other neural cells in the olfactory bulb. Transport from the cell body to the synapses of the axons (which are in the CNS) requires anterograde transport, not retrograde.

Line 112: Virus titers of Omicron BA.1 are around 2-3 log lower than delta and gamma, but not significant? How is this calculated?

Line 125- 131: Concerning the assessment of olfaction, how can it be excluded that 'sick' hamsters are not interested in food, even after 24 hours of fasting. Is there a control group where infected hamsters are given cereals after 24 hours of fasting to ensure that they will eat/collect these. Without this control, it is not possible to exclude that not finding the cereal (as in fig1G) is related to the sickness of the hamsters.

Line 151-159: Where nasal swabs collected on a daily basis to know if the differences in viral titers are 4 dpi is a result of differences in kinetics or higher/lower levels of virus replication.

Line 180-181: The data do not provide evidence for virus replication in the olfactory bulbs of the hamster, but it cannot be exclude that this occurs below the detection limit.

Line 203-205: Please include comparative data in Figure 3A from (recombinant) wild type viruses to draw this conclusion.

Line 252: Several studies have shown infection of neurons in vitro (PMID: 34031650, PMID: 34160239)

Line 251-265: If this is a model to show if SARS-CoV-2 can travel transaxonally in olfactory sensory neurons, the design is not correct. As mentioned above, virus would need to travel anterograde instead of retrograde.

Line 279: olfactory nerve instead of olfactory bulb.

Line 288-289. The authors state the that neuroinvasiveness, neurotropism and neurovirulence of SARS-CoV-2 is similar as that from other variants. However, although often not significant (most likely due to the small number of animals) there is a consistent trend that there is less virus in the olfactory bulbs (fig 2A, 2B), less abundant expression of CXCL10 & IL-10 (Fig 2E). So, I would argue that Omicron still has a neuroinvasive, neurotropic and neurovirulent potential, but that it might be reduced compared to the Wuhan strain, and other earlier variants.

Reviewer #3 (Remarks to the Author):

In this work combination of in vivo, in vitro and imaging experiments have been applied to compare SRAS-CoV-2 and its variants for their effect on anosmia and neuroinvasion in golden hamsters. Authors distinguish clinical manifestations of the diseases between original SARS-CoV-2 (Wuhan) and variants (gamma, delta and omicron). Olfactory performance has been used as an index of anosmia and viral titer, viral RNA load and expression of specific immune mediators in olfactory bulb and brain were used as an index of neuroinvasion. The results are further validated by light-sheet imaging of olfactory bulb, and in vivo and ex vivo bioluminescence imaging of the recombinant SARS-Cov-2/nLuc and Delta variant (Delta-nLuc). Authors show that regardless of the severity of the clinical signs and anosmia in the Wuhan group compared to variants, expression of immune mediators in the olfactory bulb seems insignificant, by which authors conclude that neuroinvasion and anosmia are independent process. This is later approved by testing anosmia and clinical profile in ORF7-deleted mutant of SARS-CoV-2. Using imaging and in vitro culturing in the two-compartment microfluidic devices author claim that the main entry point of the SARS-CoV-2 towards brain is the retrograde transmission of viral particles through axons. The approach and results are novel, however some concerns which needs to be addressed:

- The results on other parts of the brain are not sufficient. Except than bioluminescent images that show very limited signal in other parts of the brain. For instance, tissue inflammatory data of the rest of brain. Otherwise, authors need to address the neuroinvasion only in the olfactory bulb and not as general neuroinvasion to whole brain. This is also in contradiction with considering olfactory bulb as the main root of viral invasion to brain while the other parts of brain, or at least in the limited time period of study of 4 dpi, seems that have not been affected.
- In Figure 2F Image from ventral view of brain shows expansion outside of the olfactory bulb but in the supplementary image 4E it was not shown.
- Bioluminescent images from other groups except than Delta_nLuc have not been presented. Even though a fluorescent image from ORF7-deleted mutant is shown in Figure 4G, it might be helpful to have a comparable bioluminescent image from at least ORF7-deleted group regarding neuroinvasion.
- Olfactory performance in each group has been represented as percentage of failed animals at specific time points. Have authors tried to compare the animals of same group that show different olfactory performances? This is particularly important in Wuhan and ORF-7 mutant groups to check if there is correlation between immune response and olfactory performance in animals of the same group.
- A clear improvement in olfactory performance of ORF-7 mutant group compared to Wuhan needs to be supported by gene expression for immune mediators, bioluminescence imaging and as well with in vitro test in microfluidic device for ORF-7 mutant. This data is needed to show if anosmia in ORF-7 mutant group is not dependent to neuroinvasion.
- The opening of the microchannels at their end is about 3 μm while the size of the viral particles is in nm scales authors should confirm the lack of diffusion-related contamination in control devices without cultured neurons.

- In vitro experiment in microfluidic device looks incomplete and needs to be supported by statistical data (a measure of viral particles that have been appeared in the source reservoir), and including other groups (at least in delta variant and ORF-7 mutant). That would be interesting to see if delta variant and ORF-7 mutant show the same or different rates of retrograde transport. PCA analysis of tissue related inflammation can also include the ORF-7 mutant to see if the phenotype of this group is similar or different from variants.

Point-by-point answers to the Reviewers

Reviewer #1 (Remarks to the Author):

The presented study lead by Larrous and Bourhy focus on neuroinvasion and inflammatory response of the olfactory bulb and nasal mucosa in hamsters infected with different SARS-CoV-2 strains. The authors characterized clinically the disease course, measured TCID50, viral RNA (including subgenomic RNA) in the specific regions and show that despite different clinical signs (including olfaction) and lung pathology in the different strains, the olfactory bulb showed in all genomic viral RNA. And virus detection via nanoluciferase SARS-CoV2 signal in animals with Delta or Wuhan strains infected. This is interpreted as infection of the olfactory bulb via the epithelium as the authors demonstrated that SARS-CoV-2 travels retrogradely along axons in microfluidic chambers in vitro.

Overall the study is complementary to previous data focusing of olfactory epithelium (Chen et al. biorxiv 2022) or inflammatory brain response (Frere et al. 2022). The experimental design is solid, however I have some concerns about interpretation of the data and technical questions:

1.) Data interpretation: The authors show convincingly that SARS-CoV-2 can be transported via axons in vitro, and that virus can be found in the olfactory bulb in infected animals. However, the authors do not show neuroinflammatory reactions that one would expect during viral CNS infection as microglial nodules/activated microglial, infiltrating T cells, dead neurons as in neurotropic virus. So, if SARS-CoV-2 is “infecting” neurons, this is probably non-reproductive, and not leading to classical neuroinflammation nor functional alterations as animals infected with Omicron/Delta do not show altered olfaction despite similar viral RNA detection in the tissue. The question is if the detected RNA/ luciferase signal is at all a sign of infection. Is SARS-CoV-2 on this model neurotropic?

R.: We agree that one would expect to observe glial activation, infiltration of lymphocytes, etc following the infection of a neurotropic virus. However, some neurotropic pathogens have evolved to invade the brain unnoticed: rabies virus for example, induces only mild histopathological changes in the brain, and only in late phases after brain infection (de Melo et al., 2020, Fooks et al., 2017). We are not assuming that SARS-CoV-2 behaves similarly as rabies virus, but, it is possible for a pathogen to be detected in the CNS without major histopathological alterations.

More specifically to SARS-CoV-2, in a previous study with the Wuhan virus (de Melo et al., 2021a), we did not observe tissue reaction in infected brain regions. We could isolate the virus from different brain regions (olfactory bulbs, brainstem, cortex, cerebellum). These same regions were also positive for viral RNA (RT-qPCR), however, by immune-fluorescence in brain sections, we only detected few tissular reaction in the olfactory bulbs, probably due to the very restricted infection area.

cell. (E) Sagittal section showing nasal turbinates and olfactory bulb of SARS-CoV-2-infected hamster at 4 dpi. Inset depicts SARS-CoV-2 staining in OSN axons. (F) Olfactory sensory axons projecting into glomeruli in the olfactory bulb of SARS-CoV-2-inoculated hamsters at 4 dpi. Insets (F' and F'') show infected cells. (G to I) Olfactory bulb of mock-infected (G) or SARS-CoV-2-infected (H and I) hamsters at 4 dpi. Iba1⁺ infected cells are shown in (H), and several infected cells are observed in (I). SARS-CoV-2 is detected by antibodies raised against the viral NP. Scale bars, 20 μm (A to D and H), 100 μm (E and F), and 50 μm (G and I). Images are single z planes (A to H) or maximum intensity projection over a 6-μm depth (I). (J) Number of NP⁺

de Melo GD, Parize P, Jouvion G, Dacheux L, Chrétien F, Bourhy H. Rabies. *Infections of the Central Nervous System* 2020. p. 121-9.

Fooks AR, Cliquet F, Finke S, Freuling C, Hemachudha T, Mani RS, Müller T, Nadin-Davis S, Picard-Meyer E, Wilde H, Banyard AC.

Rabies. Nature Reviews Disease Primers. 2017;3(1):17091.

de Melo GD, Lazarini F, Levallois S, Hautefort C, Michel V, Larrous F, et al. COVID-19-related anosmia is associated with viral persistence and inflammation in human olfactory epithelium and brain infection in hamsters. *Science Translational Medicine. 2021;13(596):eabf8396.*

Regarding the question “The question is if the detected RNA/ luciferase signal is at all a sign of infection.”, viral RNA does not mean viral infection, but that viral RNA is present in the tissue; on the contrary, luciferase signal is a sign of infection as the production of luciferase is linked to active viral mRNA transcription. This was completed by the viral titer in the tissues, which indicates the liberation of infectious viral particles.

Regarding the question “Is SARS-CoV-2 on this model neurotropic?”, we show herein that SARS-CoV-2 can enter the neuron and travel along axons; that it can infect the olfactory bulbs, and that it can elicit an inflammatory response within the CNS. All these points give support to the hypothesis of the neurotropic potential of SARS-CoV-2.

Furthermore, “No virus was observed in other areas of the brain”. But this data is not shown in the manuscript, except for luciferase dataset. But is there RNA detectable? Protein? Is the inflammatory change the authors describe also detectable in areas without viral RNA/protein? This would point to a more general bystander reaction instead of a direct virus mediated effect. And to address the neuroinflammation: Are there signs of neuroinflammation in the olfactory bulb, this is not clear after measuring IFN β , MX2 etc.? Cellular changes should be analyzed as reactivated microglia, infiltrating T cells etc.

R.: We agree that this point was not clear. The sentence was rewritten (lines 218-220) to highlight that, no infected cells were observed in other areas using IHC and light sheet imaging (iDISCO+). In this manuscript we focused on the olfactory bulbs only. Following a previous article from our group (de Melo et al, 2021a), we have obtained positive viral titers and positive qPCRs in other brain areas (brainstem, cerebral cortex, cerebellum), but again, the visualization of positive viral staining was only detected in the olfactory bulbs, with very low cellular reaction (microglia). Based on the same article, we have selected the most relevant genes that indicate a host tissue response to the infection (Mx2, Ifn-beta, Il-6, Cxcl10, Tnf-alpha, and Il-10).

de Melo GD, Lazarini F, Levallois S, Hautefort C, Michel V, Larrous F, et al. COVID-19-related anosmia is associated with viral persistence and inflammation in human olfactory epithelium and brain infection in hamsters. Science Translational Medicine. 2021;13(596):eabf8396.

2.) Technical issues:

- the reader should be enabled to connect clinical picture and molecular/imaging data better, therefore animals could e.g. be identifiable with a code and this information should be given in the respective figure legends together with a table with clinical data values. E.g. one would like to check as a reader if the animals detected with luciferase signal in the olfactory bulb had also impaired olfaction or not. And similar viral protein in the bulb is only shown for Wuhan strain via light sheet of one animal?

R.: We think that adding new codes/colors would render the graphs too heavy and would impact the reading. Therefore, we added an excel file containing the source data of the figures.

Light sheet imaging was only performed for Wuhan-infected animals (3 males and 3 females) as an attempt to visualize infected cells. We updated this information on the reporting summary.

-As the number of animals is in some of the experiments very limited (probably due to animal protection reasons) I recommend that the authors show all data (all animals analyzed) and not only "representative" images especially in the lightsheet dataset.

R.: New figures were added as well as an excel file containing the source data of the figures.

- What is the rationale of the genes selected for inflammatory reaction?

R.: Based on our previous article (de Melo et al., 2021a), we have selected the most relevant genes that indicate a host tissue response to the infection (Mx2, Ifn-beta, Il-6, Cxcl10, Tnf-alpha, and Il-10).

de Melo GD, Lazarini F, Levallois S, Hautefort C, Michel V, Larrous F, et al. COVID-19-related anosmia is associated with viral persistence and inflammation in human olfactory epithelium and brain infection in hamsters. Science Translational Medicine. 2021;13(596):eabf8396.

-the method section is lacking important information regarding fluorescent microscopy of the in vitro part; an excellent guideline regarding this topic is given in Llopis et al. Nature methods 2021.

R.: We completed the information in lines 787-788.

Minor points:

-in the discussion the authors list studies that supposedly show brain infection with SARS-CoV-2. These studies show indeed viral RNA in different brain regions but no sgRNA and no viral particles and the spike staining is not convincing due to high "background" staining also in controls as reported e.g., by Yang et al. Nature 2021...as viral RNA is not equal brain infection, brain invasion of SARS-CoV-2 in humans has not been shown so far up to my knowledge.

R.: We thank the reviewer to raise this point of discussion. We have indeed included a list of studies that report brain infection, we have also added a list of studies that did not detect brain infection (lines 287-289): *While studies on natural infection in humans and experimental infection in animal models reported brain infection by SARS-CoV-2^{13,30-33}, including in in vitro human models of infection^{34,35}, others failed to detect SARS-CoV-2 in the nervous tissue^{5,36}.* We completed this rationale in lines 284-287).

We fully agree that viral RNA is not equal brain infection, and that's why we used many different molecular, virologic and imaging approaches to validate our findings, either by direct isolation of infectious viral particles (Figure 2A), either by detection of markers of viral replication (production of nanoluciferase) or by detecting infected cells (Figure 3).

-for the given clone of nucleocapsid the company doesn't show a tissue staining, how did the authors validate/verify antibody specificity and sensitivity?

R.: Actually, the company report at least 19 publications using this antibody against SARS-CoV-2 (https://www.novusbio.com/products/sars-nucleocapsid-protein-antibody_nb100-56576#reviews-publications). Further, we and others also used this antibody in other studies:

Frantz PN, Barinov A, Ruffié C, Combredet C, Najburg V, de Melo GD, et al. A live measles-vectored COVID-19 vaccine induces strong immunity and protection from SARS-CoV-2 challenge in mice and hamsters. Nature Commun. 2021;12(1):6277.

Martines RB, Ritter JM, Matkovic E, et al. Pathology and Pathogenesis of SARS-CoV-2 Associated with Fatal Coronavirus Disease, United States. Emerging Infectious Diseases. 2020;26(9):2005-2015. doi:10.3201/eid2609.202095.

-it is not clear for me, if only male hamsters were used as written in the reporting summary or also female as written in the methods section, here again an overview with all included animals with respective analysis would help. Were the female hamsters only used for light-sheet analysis?

R.: As female hamsters present a very low rate of olfaction loss (de Melo et al., 2021b), we decided not to use females in this study, which is based on the induction of anosmia by different SARS-CoV-2 VoCs. However, exceptionally in the light-sheet imaging, we included male and female hamsters. We updated this information on the reporting summary.

de Melo GD, Lazarini F, Larrous F, Feige L, Kornobis E, Levallois S, et al. Attenuation of clinical and immunological outcomes during SARS-CoV-2 infection by ivermectin. EMBO Mol Med. 2021;13:e14122.

- video S2 needs more explanation please, what is the red structure?

R.: We replace the videos S1 and S2 by a new version of video S1 containing embedded explanations (lines 515-518).

-How was PCA computation done? How were PCA plots generated?

R.: The PCA analysis was performed using GraphPad Prism v.9 as stated in the statistics section (lines 795-801).

-Sometimes the authors use nucleocapsid (correct) but sometimes also nucleoprotein, this should be adapted.

R.: We replaced nucleoprotein to nucleocapsid throughout the text.

-Single channel images for all shown fluorescent images should be made available in the supplement or via online depository.

R.: We added single channel images in Supplementary Figures 6 and 7.

Reviewer #2 (Remarks to the Author):

De manuscript by Dias de Melo et al. is an elegant study which combines different techniques to monitor SARS-CoV-2 infection in hamsters, and to visualize virus invasion into the CNS. This study reveals that all SARS-CoV-2 variants can invade the CNS via the olfactory nerve, independent of their capacity to cause disease, including anosmia. However, due to the limited numbers of animals, it is difficult to determine whether there are differences among the SARS-CoV-2 variants included in the studies. In addition, I have some concerns whether the assessment of olfaction includes all necessary controls.

Specific comments:

Introduction: It is not introduced why the authors included SARS-CoV-2 with a ORF7ab deletion in their studies.

R.: We have introduced the ORF7 in lines 93-97.

Line 104 (and 322): The olfactory nerve is different from the other cranial nerves. The cell body of the olfactory sensory neuron resides in the nasal mucosa and the axon penetrates the cribriform plate where it has contact with other neural cells in the olfactory bulb. Transport from the cell body to the synapses of the axons (which are in the CNS) requires anterograde transport, not retrograde.

R.: We thank the reviewer for this warning, and we agree that our data on the retrograde transport was not adapted to validate the hypothesis of invasion via the olfactory route. We now show experiment data obtained in microfluidic devices, in a way to have all viruses (Wuhan, Wuhan/ Δ ORF7ab, Gamma, Delta, and Omicron/BA.1) in both retrograde and anterograde directions (Figure 4 and Supplementary Figures 6-7).

We completed equally the retrograde blocking study with ciliobrevin D for all viruses (Supplementary Figure 8), but we were not able to block the anterograde transport. We tried to validate a blocker for the anterograde transport using rabies virus and ciliobrevin D (inhibitor of dynein, that could be involved in the bidirectional axonal transport), monastrol (inhibitor of kinesin-5) or colchicine (induces microtubule disassembly), but without success.

Line 112: Virus titers of Omicron BA.1 are around 2-3 log lower than delta and gamma, but not significant? How is this calculated?

R.: As we stated in the Statistics section (lines 795-801), the difference between groups were calculated using the Kruskal-Wallis test followed by the Dunn's multiple comparisons test. In this version, using the Dunn's multiple comparisons test, we compared all the infected groups versus the mock, to increase power by reducing the number of pairwise comparisons. With these non-parametric analyses, even if the groups are 'visually' different, the lack of statistical difference may be a consequence of the power of this non-parametrical test or of the sample size.

Line 125- 131: Concerning the assessment of olfaction, how can it be excluded that 'sick' hamsters are not interested in food, even after 24 hours of fasting. Is there a control group where infected hamsters are given cereals after 24 hours of fasting to ensure that they will eat/collect these.

Without this control, it is not possible to exclude that not finding the cereal (as in fig1G) is related to the sickness of the hamsters.

R.: The olfaction tests were performed as we have previously established (de Melo et al., 2021a), based on a first trial with hidden food, and a subsequent trial in the same conditions, but with visible food. All animals found and ate the visible food. We added this information in lines 132-134 and we also complemented the olfactory test description with the visible food in lines 642-643.

de Melo GD, Lazarini F, Levallois S, Hautefort C, Michel V, Larrous F, et al. COVID-19-related anosmia is associated with viral persistence and inflammation in human olfactory epithelium and brain infection in hamsters. Science Translational Medicine. 2021;13(596):eabf8396.

Line 151-159: Where nasal swabs collected on a daily basis to know if the differences in viral titers are 4 dpi is a result of differences in kinetics or higher/lower levels of virus replication.

R.: This could provide interesting information, but collection of nasal swabs is not feasible *intra vitam* in hamsters.

Line 180-181: The data do not provide evidence for virus replication in the olfactory bulbs of the hamster, but it cannot be exclude that this occurs below the detection limit.

R.: We agree with this point and changed the sentence (line 200).

Line 203-205: Please include comparative data in Figure 3A from (recombinant) wild type viruses to draw this conclusion.

R.: We added the data of Wuhan and Delta original viruses in Figure 3C as requested.

Line 252: Several studies have shown infection of neurons in vitro (PMID: 34031650, PMID: 34160239)

R.: We have rewritten the sentence (line 257-258).

Line 251-265: If this is a model to show if SARS-CoV-2 can travel transaxonally in olfactory sensory neurons, the design is not correct. As mentioned above, virus would need to travel anterograde instead of retrograde.

R.: In this version of the manuscript, we now included data from experiments performed in microfluidic devices also considering the anterograde direction (Figure 4 and Supplementary Figures 6-7).

Line 279: olfactory nerve instead of olfactory bulb.

R.: The sentence was corrected (line 296).

Line 288-289. The authors state that the neuroinvasiveness, neurotropism and neurovirulence of SARS-CoV-2 is similar as that from other variants. However, although often not significant (most likely due to the small number of animals) there is a consistent trend that there is less virus in the olfactory bulb (fig 2A, 2B), less abundant expression of CXCL10 & IL-10 (Fig 2E). So, I would argue that Omicron still has a neuroinvasive, neurotropic and neurovirulent potential, but that it might be reduced compared to the Wuhan strain, and other earlier variants.

R.: We totally agree with this point, and we rewritten this interpretation in lines 303-305.

Reviewer #3 (Remarks to the Author):

In this work combination of in vivo, in vitro and imaging experiments have been applied to compare SARS-CoV-2 and its variants for their effect on anosmia and neuroinvasion in golden hamsters. Authors distinguish clinical manifestations of the diseases between original SARS-CoV-2 (Wuhan) and variants (gamma, delta and omicron). Olfactory performance has been used as an index of anosmia and viral titer, viral RNA load and expression of specific immune mediators in olfactory bulb and brain were used as an index of neuroinvasion. The results are further validated by light-sheet imaging of olfactory bulb, and in vivo and ex vivo bioluminescence imaging of the recombinant SARS-CoV-2/nLuc and Delta variant (Delta-nLuc). Authors show that regardless of the severity of the clinical signs and anosmia in the Wuhan group compared to variants, expression of immune mediators in the olfactory bulb seems insignificant, by which authors conclude that neuroinvasion and anosmia are independent processes. This is later approved by testing anosmia and clinical profile in ORF7-deleted mutant of SARS-CoV-2. Using imaging and in vitro culturing in the two-compartment microfluidic devices authors claim that the main entry point of the SARS-CoV-2 towards brain is the retrograde transmission of viral particles through axons. The approach and results are novel, however some concerns which need to be addressed:

- The results on other parts of the brain are not sufficient. Except than bioluminescent images that show very limited signal in other parts of the brain. For instance, tissue inflammatory data of the rest of brain. Otherwise, authors need to address the neuroinvasion only in the olfactory bulb and not as general neuroinvasion to whole brain. This is also in contradiction with considering olfactory bulb as the main route of viral invasion to brain while the other parts of brain, or at least in the limited time period of study of 4 dpi, seems that have not been affected.

R.: Even if there are evidence of SARS-CoV-2 infecting other brain areas (de Melo et al., 2021a), in this study we specifically focused on the olfactory bulbs to relate it to anosmia. Regarding the time-points, 4 days post-infection is not limited considering the hamster model; this time-point is the peak of the acute phase, when the viral load is high, the animals are symptomatic and before the clinical healing.

de Melo GD, Lazarini F, Levallois S, Hautefort C, Michel V, Larrous F, et al. COVID-19-related anosmia is associated with viral persistence and inflammation in human olfactory epithelium and brain infection in hamsters. Science Translational Medicine. 2021;13(596):eabf8396.

Chan JF, Zhang AJ, Yuan S, Poon VK, Chan CC, Lee AC, et al. Simulation of the clinical and pathological manifestations of Coronavirus Disease 2019 (COVID-19) in golden Syrian hamster model: implications for disease pathogenesis and transmissibility. Clin Infect Dis. 2020.

Sia SF, Yan LM, Chin AWH, Fung K, Choy KT, Wong AYL, et al. Pathogenesis and transmission of SARS-CoV-2 in golden hamsters. Nature. 2020;583(7818):834-8.

- In Figure 2F Image from ventral view of brain shows expansion outside of the olfactory bulb but in the supplementary image 4E it was not shown.

R.: Regarding in vivo imaging, the color code is for visualization only. The comparison factor is the radiance obtained in a determined region of interested (ROI). Nevertheless, we observed some discrepancies in the color scale between the Figure 3 and Supplementary Figure 4. In the current version, after harmonization, the lungs in both figures are represented in the same color scale (min 2×10^7 - max 2×10^8), as well as the brains (min 2×10^8 - max 5×10^8).

- Bioluminescent images from other groups except than Delta_nLuc have not been presented. Even though a fluorescent image from ORF7-deleted mutant is shown in Figure 4G, it might be helpful to have a comparable bioluminescent image from at least ORF7-deleted group regarding neuroinvasion.

R.: After this suggestion, we were successful in the production of a new recombinant Wuhan/ Δ ORF7ab virus expressing the nLuc (Wuhan/ Δ ORF7ab_nLuc), and now, the Figure 3 contains data from three different viruses: Wuhan_nLuc, Wuhan/ Δ ORF7ab_nLuc, and Delta_nLuc.

- Olfactory performance in each group has been represented as percentage of failed animals at specific time points. Have authors tried to compare the animals of same group that show different olfactory performances? This is particularly important in Wuhan and ORF-7 mutant groups to check if there is correlation between immune response and olfactory performance in animals of the same group.

R.: We have indeed tried to find signatures related to olfaction loss, including by PCA (Supplementary Figure 3), but without success, since inflammation seems not to be related to anosmia, at least according to the parameters analyzed in this study. To facilitate individual analysis of the animals, we added an excel file containing the source date of the figures.

- A clear improvement in olfactory performance of ORF-7 mutant group compared to Wuhan needs to be supported by gene expression for immune mediators, bioluminescence imaging and as well with in vitro test in microfluidic device for ORF-7 mutant. This data is needed to show if anosmia in ORF-7 mutant group is not dependent to neuroinvasion.

R.: The data regarding Wuhan/ Δ ORF7ab were added in Figures 1-2, and the data regarding Wuhan/ Δ ORF7ab_nLuc were included in Figure 3.

- The opening of the microchannels at their end is about 3 μ m while the size of the viral particles is in

nm scales authors should confirm the lack of diffusion-related contamination in control devices without cultured neurons.

R.: The isolation of the two connected culture chambers is also assured by the fluid dynamics. To prevent passive diffusion of assemblies, hydrostatic pressure was applied by adding medium excess in the reservoirs of recipient chambers. We added this information in lines 774-776. Further, the absence of passive diffusion can also be observed in the experiments with dynein inhibitors, where no infected cells were observed in the left chamber (Supplementary Figure 8).

- In vitro experiment in microfluidic device looks incomplete and needs to be supported by statistical data (a measure of viral particles that have been appeared in the source reservoir), and including other groups (at least in delta variant and ORF-7 mutant). That would be interesting to see if delta variant and ORF-7 mutant show the same or different rates of retrograde transport.

R.: We completed the microfluidic study to include all viruses (Wuhan, Wuhan/ Δ ORF7ab, Gamma, Delta, and Omicron/BA.1) in both retrograde and anterograde directions (Figure 4 and Supplementary Figures 6-7). To analyze a measurement of the viral particles arriving in the non-infected wells, we used the nLuc recombinant viruses (Wuhan_nLuc, Wuhan/ Δ ORF7ab_nLuc, and Delta_nLuc) that allowed a follow-up measurement of the viral load via the nLuc activity in the supernatants (Supplementary Figure 9).

- PCA analysis of tissue related inflammation can also include the ORF-7 mutant to see if the phenotype of this group is similar or different from variants.

R.: The PCA analysis was completed with the Wuhan/ Δ ORF7ab data (Supplementary Figure 3).

REVIEWER COMMENTS

Reviewer #1 (Remarks to the Author):

The authors explained, added or corrected almost all my concerns or misunderstandings. The manuscript is improved

The only issue not sufficiently clarified yet is the validation of the antibody as the company is not providing a validation on the website. Monya Baker wrote a nice feature on this issue and why this can be dangerous. <https://www.nature.com/articles/d41586-020-02549-1>

I will explain in more detail why I think in this context the given publications are not sufficient:

- 1.) The reporting summary of Frantz et al. does not mention the antibody and in the method section no further information could be found by me.
- 2.) In Martines et al. the authors describe a test against crossreactivity in infected (human I guess) tissue samples but do not describe how they validated sensitivity, reproducibility etc. The citation given here in the method section is linked to a publication which used another antibody.
- 3.) In the publication from science transl medicine (from the authors and which they are referring to in the reporting summary regarding validation of all antibodies) I could not find the information that the same antibody was used as it is written there: "rabbit anti-SARS-CoV NP (1/500; provided by N. Escriou, Institut Pasteur, Paris." And not data on validation could be found by me.

This is why I think the authors should please mention a paper that specifically shows validation in a similar context as they use it, or show the respective data in this manuscript. Or maybe I just could not find it, in the given citations, then the authors could please help me to extract the validation information already published.

Reviewer #2 (Remarks to the Author):

The majority of my comments (Reviewer #2) have been adequately addressed. However, the part of the studies that describe retrograde and anterograde transport of SARS-CoV-2 variants in the microfluidic chambers are not clearly described.

Both in the results sections (line 256 onwards) and materials and methods section (line 751 onwards) describe that in the microfluidic chambers the neuron-epithelial cultures are seeded in both the left and right chamber. However, that is not how the networks are shown in the Figure 4. In Figure 4 neurons are only seeded in the left chamber, which would allow studying retrograde and anterograde transport along axons by adding virus on the right or left compartment retrospectively. However, if neurons are present in both the left and right compartment how is it possible to exclusively study retrograde or anterograde transport in this model? In order to study anterograde transport the compartment with the

soma needs to be inoculated, after which axonal transport (and release) to the axon termini needs to be analyzed (as represented in Figure 4).

Concerning this comment:

Line 252: Several studies have shown infection of neurons in vitro (PMID: 34031650, PMID: 34160239)

R.: We have rewritten the sentence (line 257-258)

Response reviewer #2: Indeed the sentence is rewritten, but it still suggests that co-cultures with permissive cells are necessary, which is not the case. Induced pluripotent stem cell derived neurons are susceptible—but not always permissive—for SARS-CoV-2, which might also depend on the type of neuron and/or differentiation method.

Reviewer #3 (Remarks to the Author):

All concerns have been addressed properly by authors that include incorporation of new data and statistical analysis into the manuscript and new images.

Point-by-point answers to the Reviewers

Reviewer #1 (Remarks to the Author):

The authors explained, added or corrected almost all my concerns or misunderstandings. The manuscript is improved. The only issue not sufficiently clarified yet is the validation of the antibody as the company is not providing a validation on the website. Monya Baker wrote a nice feature on this issue and why this can be dangerous. <https://www.nature.com/articles/d41586-020-02549-1>

I will explain in more detail why I think in this context the given publications are not sufficient:

- 1.) The reporting summary of Frantz et al. does not mention the antibody and in the method section no further information could be found by me.
- 2.) In Martines et al. the authors describe a test against crossreactivity in infected (human I guess) tissue samples but do not describe how they validated sensitivity, reproducibility etc. The citation given here in the method section is linked to a publication which used another antibody.
- 3.) In the publication from science transl medicine (from the authors and which they are referring to in the reporting summary regarding validation of all antibodies) I could not find the information that the same antibody was used as it is written there: "rabbit anti-SARS-CoV NP (1/500; provided by N. Escriou, Institut Pasteur, Paris." And not data on validation could be found by me.

This is why I think the authors should please mention a paper that specifically shows validation in a similar context as they use it, or show the respective data in this manuscript. Or maybe I just could not find it, in the given citations, then the authors could please help me to extract the validation information already published.

R.: We thank the reviewer for insisting in this point as the info was not clear.

A validation statement for this antibody is already published in Munster et al. (2020), in both Methods section and in the respective reporting summary:

Munster VJ, Feldmann F, Williamson BN, van Doremalen N, Pérez-Pérez L, Schulz J, Meade-White K, Okumura A, Callison J, Brumbaugh B, Avanzato VA, Rosenke R, Hanley PW, Saturday G, Scott D, Fischer ER, de Wit E. Respiratory disease in rhesus macaques inoculated with SARS-CoV-2. *Nature*. 2020;585(7824):268-72.

For more precision:

- 1) Frantz et al.: the antibody is presented in page 14 under the Methods section "Golden Syrian hamsters immunization and challenge" but indeed, it is not mentioned in the reporting summary.
- 2) Martines et al.: this article is one of the 21 articles indicated by the supplier as references for this antibody and SARS-CoV-2 (https://www.novusbio.com/products/sars-nucleocapsid-protein-antibody_nb100-56576#reviews-publications)
- 3) We realized that the reference in the reporting summary was not the good one, therefore we added Munster et al. (2020).

Reviewer #2 (Remarks to the Author):

The majority of my comments (Reviewer #2) have been adequately addressed. However, the part of the studies that describe retrograde and anterograde transport of SARS-CoV-2 variants in the microfluidic chambers are not clearly described.

Both in the results sections (line 256 onwards) and materials and methods section (line 751 onwards) describe that in the microfluidic chambers the neuron-epithelial cultures are seeded in both the left

and right chamber. However, that is not how the networks are shown in the Figure 4. In Figure 4 neurons are only seeded in the left chamber, which would allow studying retrograde and anterograde transport along axons by adding virus on the right or left compartment retrospectively. However, if neurons are present in both the left and right compartment how is it possible to exclusively study retrograde or anterograde transport in this model? In order to study anterograde transport the compartment with the soma needs to be inoculated, after which axonal transport (and release) to the axon termini needs to be analyzed (as represented in Figure 4).

R.: We redraw the scheme in Figure 4A to better explain the culture system in the microfluidic chambers, as neurons and epithelial cells were seeded in both chambers. The anterograde and retrograde transport can be assessed in these cultures because the chambers are connected by the axons from neurons whose soma are in the left chamber. The microchannels allow the axonal growth from the left to the right chamber only, due to their funnel-shape structure. We updated the legend of Figure 4 (lines 408-411) and the figures S6-S9.

Concerning this comment:

Line 252: Several studies have shown infection of neurons in vitro (PMID: 34031650, PMID: 34160239)

R.: We have rewritten the sentence (line 257-258)

Response reviewer #2: Indeed the sentence is rewritten, but it still suggests that co-cultures with permissive cells are necessary, which is not the case. Induced pluripotent stem cell derived neurons are susceptible—but not always permissive—for SARS-CoV-2, which might also depend on the type of neuron and/or differentiation method.

R.: We added more information in this paragraph to bring the matter for discussion and to state that with our protocol, we were unable to infect neurons when cultured alone (lines 257-262).

Reviewer #3 (Remarks to the Author):

All concerns have been addressed properly by authors that include incorporation of new data and statistical analysis into the manuscript and new images.

R.: We thank the reviewer for this positive evaluation.

REVIEWERS' COMMENTS

Reviewer #2 (Remarks to the Author):

My concerns have been addressed and/or clarified. I do not have any additional concerns.